# Inter-continental variability in the relationship of oxidative potential and cytotoxicity with PM$_{2.5}$ mass

Sudheer Salana [1], Haoran Yu[1,2], Zhuying Dai[1], P. S. Ganesh Subramanian [1], Joseph V. Puthussery [1,3], Yixiang Wang[1,4], Ajit Singh [5,6], Francis D. Pope [5], Manuel A. Leiva G.[7], Neeraj Rastogi [8], Sachchida Nand Tripathi [9,10], Rodney J. Weber [11] & Vishal Verma [1] ✉

Most fine ambient particulate matter (PM$_{2.5}$)-based epidemiological models use globalized concentration-response (CR) functions assuming that the toxicity of PM$_{2.5}$ is solely mass-dependent without considering its chemical composition. Although oxidative potential (OP) has emerged as an alternate metric of PM$_{2.5}$ toxicity, the association between PM$_{2.5}$ mass and OP on a large spatial extent has not been investigated. In this study, we evaluate this relationship using 385 PM$_{2.5}$ samples collected from 14 different sites across 4 different continents and using 5 different OP (and cytotoxicity) endpoints. Our results show that the relationship between PM$_{2.5}$ mass vs. OP (and cytotoxicity) is largely non-linear due to significant differences in the intrinsic toxicity, resulting from a spatially heterogeneous chemical composition of PM$_{2.5}$. These results emphasize the need to develop localized CR functions incorporating other measures of PM$_{2.5}$ properties (e.g., OP) to better predict the PM$_{2.5}$-attributed health burdens.

Air quality policy measures are largely dictated by epidemiological studies. In these studies, fine ambient particulate matter (i.e., particles below 2.5 μm in aerodynamic diameter, called PM$_{2.5}$ hereafter)-induced mortality due to various causes such as stroke, lung cancer, or lower respiratory infection (LRI) is often estimated using globalized concentration-response (CR) functions[1,2]. These CR relationships have been constructed using relative risk (RR) estimates from cohort studies conducted in limited regions, mostly in North America, Europe, and China[3–5], and/or using the RR estimates from certain PM$_{2.5}$ sources such as solid cooking fuel, second-hand tobacco

smoke, and active smoking to account for higher concentration exposure[1,3,6]. Accordingly, these CR functions neither cover the entire range of sources of ambient PM$_{2.5}$, nor account for the spatio-temporal variations in its chemical composition which varies widely across the world[7,8]. Moreover, globalized CR functions do not account for the variations in health responses of the individuals living in different geographical regions, owing to their different physiological, climatic, and social backgrounds. Therefore, studies estimating global mortality risk from ambient PM$_{2.5}$ based on these CR functions, such as Global Burden of Disease report[9] include an

[1]Department of Civil and Environmental Engineering, University of Illinois at Urbana Champaign, Urbana, IL 61801, USA. [2]Department of Civil and Environmental Engineering, University of Alberta, Edmonton, AB, Canada. [3]Department of Energy, Center for Aerosol Science and Engineering, Environmental and Chemical Engineering, Washington University in St. Louis, St. Louis, MO 63130, USA. [4]College of Health, Lehigh University, Bethlehem, PA 18015, USA. [5]School of Geography, Earth and Environmental Sciences, University of Birmingham, Birmingham B15 2TT, UK. [6]Institute of Applied Health Research, University of Birmingham, Edgbaston, Birmingham B15 2TT, UK. [7]Department of Chemistry, Faculty of Science, Universidad de Chile, Las Palmeras 3425, Ñuñoa, Santiago, RM, Chile. [8]Geosciences Division, Physical Research Laboratory, Ahmedabad 380009, India. [9]Department of Civil Engineering, Indian Institute of Technology Kanpur, Kanpur 208016, India. [10]Department of Sustainable Energy Engineering, Indian Institute of Technology Kanpur, Kanpur 208016, India. [11]School of Earth and Atmospheric Sciences, Georgia Institute of Technology, Atlanta, GA 30332, USA. ✉e-mail: vverma@illinois.edu

inherent assumption that the toxicity of $PM_{2.5}$ is a function of mass alone and is immune to different emission sources, atmospheric processing, the resulting chemical composition, and disparity in health responses of the individuals living in different geographical regions. Predictions from such studies eventually lead to policy measures which are solely focused on reducing total ambient $PM_{2.5}$ mass concentrations, while ignoring source contributions and atmospheric processing of the $PM_{2.5}$. However, whether such policy measures would yield equivalent health benefits is unclear due to two reasons. First, $PM_{2.5}$ is a mixture of several chemical components and the toxicities of these chemical species have been shown to be different[10,11]. And 2nd, there is a dearth of literature comparing the spatiotemporal variations in toxic $PM_{2.5}$ components versus $PM_{2.5}$ mass, to validate the assumption that $PM_{2.5}$ mass by itself can sufficiently capture the spatiotemporal variation in the toxicity of ambient $PM_{2.5}$. Owing to these concerns, the effectiveness of the $PM_{2.5}$-mass-based approach to establish cause and effect between air quality and health impact is questionable.

Among several routinely measured properties of the $PM_{2.5}$, determining the most health-relevant property is also a topic of great research. In the last few decades, oxidative stress, which is caused by an imbalance of reactive oxygen species (ROS) and antioxidants[12], has emerged as the underlying pathology of many diseases[13–15]. Oxidative potential (OP) is defined as the capability of $PM_{2.5}$ to induce oxidative stress, and therefore, OP has been purported as a possible proxy for $PM_{2.5}$ toxicity. Different endpoints to quantify OP have been used and these include both acellular[16,17] and cellular assays[18–20] to measure ROS generation and antioxidant depletion. While toxicological studies have linked OP with several cellular endpoints such as cytotoxicity[21,22], cellular oxidative stress[16], and inflammatory response[16,23], clinical studies have also shown association of OP with respiratory and cardiac diseases such as asthma and rhinitis[24], ischemic heart disease[25], and congestive heart failure (CHD)[26]. Several epidemiological studies have also indicated a better association of OP with adverse health endpoints such as low birth weight[27], lung cancer mortality[28], diabetes[29], emergency room visits for myocardial infarction[30], asthma, wheeze, and CHD[31], and temperature-induced cardiovascular events[32], than with $PM_{2.5}$ mass.

Although, there have been several studies in recent years which have investigated the spatiotemporal distribution of OP[18,33–35], these studies are very limited in terms of their spatial scope, often focusing on a limited number of sites in the same geographical region such as southeast US[36], Western US[37–39], Midwest US[18,33], and Europe[34], while mostly focusing on one kind of assays (i.e., either acellular or cellular). Moreover, the observed relationships between OP and $PM_{2.5}$ mass in these studies could be influenced by specific chemical composition in that region. For example, if the main drivers (i.e., $PM_{2.5}$ chemical constituents) of OP correlate well with $PM_{2.5}$ mass in a region, mass could seem to capture the $PM_{2.5}$ toxicity or OP well, but this relationship will change when OP drivers no longer correlate with mass. Therefore, the relationship between $PM_{2.5}$ mass, OP, and toxicity needs to be investigated in diverse emission scenarios and geographical settings, so that the biases from specific chemical composition could be minimized. And as of now, we are not aware of any study which has investigated these relationships on a large spatial extent. The results from such a study can be used to test the validity of the main assumption of epidemiological models mentioned earlier, i.e., if spatiotemporal variations in toxic $PM_{2.5}$ components vs. $PM_{2.5}$ mass are similar?

Our current study explores the nature of the relationship among $PM_{2.5}$ mass, OP, and cytotoxicity on a relatively large spatial extent. Here, we have used $PM_{2.5}$ samples ($N = 385$) collected from fourteen different sites across four different continents (Asia, Europe, North and South America) and evaluated five commonly used measures to surrogate for $PM_{2.5}$ toxicity [3 acellular OP endpoints – dithiothreitol

depletion ($OP^{DTT}$), glutathione depletion ($OP^{GSH}$), hydroxyl radical formation ($OP^{OH}$); 2 cellular endpoints – cytotoxicity [or cell death (CD)] using crystal violet assay and cellular OP ($OP^C$) using dichlorofluorescein diacetate (DCFH-DA) in a human lung epithelial cell line (A549)]. We then used these measurements to investigate two questions: (1) is $PM_{2.5}$ mass correlated with extrinsic OP and cytotoxicity? and if so, (2) is the association between $PM_{2.5}$ mass and extrinsic OP and cytotoxicity spatially uniform? Our study compares the responses of the most widely used OP and toxicological assays for a large number of $PM_{2.5}$ samples collected from an extensive spatial scale. In addition, the samples collected from diverse environmental settings provided us a rare opportunity to investigate the effect of substantially different $PM_{2.5}$ chemical composition as a result of uniquely different emission sources (e.g., pertinent to the specific regions) and atmospheric conditions, on intrinsic OP and toxicity, and the relationship between extrinsic OP and $PM_{2.5}$ mass. Essentially, through these measurements and comparisons, we demonstrate the need for developing localized CR functions based on the intrinsic toxicity and chemical composition of the $PM_{2.5}$ and investigating other metrics of $PM_{2.5}$ to better represent its health effects.

## Results and discussion
### Variations in $PM_{2.5}$ characteristics

Figure 1 shows the $PM_{2.5}$ mass concentrations, extrinsic OP and cytotoxicity [i.e., per $m^3$ of air; denoted as $OP^{DTT}v$, $OP^{GSH}v$, and $OP^{OH}v$ for three acellular OP endpoints, $OP^Cv$ for cellular OP, and CDv for cytotoxicity] at five geographical regions, i.e., Midwest US (average of Chicago, Bondville, Champaign, St. Louis, and Indianapolis), West Midlands, UK (average of an urban and background site in Birmingham), India (average of Ahmedabad, Hisar, Patiala, and Faridabad), Southeast US (Atlanta) and Chile (average of Santiago and Chillan), in different seasons. The corresponding data on intrinsic OP and cytotoxicity [i.e., per µg of $PM_{2.5}$; denoted as OPm and CDm] are shown in Supplementary Fig. 5 in supplementary information (SI). All the data was normalized by the MinMax scaler technique[40], which first determines the minimum ($x_{min}$) and maximum value ($x_{max}$) in a dataset, and then each value in the dataset is scaled using Eq. (1):

$$x_{\text{scaled}} = \frac{x - x_{min}}{x_{max} - x_{min}} \qquad (1)$$

where $x_{scaled}$ is the normalized value.

Thus, all the data is normalized between 0 and 1 for direct comparison among different endpoints, which were having very wide ranges. Further details of normalization methodology and the related equations are described in SI. The absolute values of the average mass, OP, and cytotoxicity (both extrinsic and intrinsic) in different seasons at these regions are shown in Supplementary Fig. 6, and Supplementary Table 4 in SI shows this dataset at all the individual sampling sites (i.e., without averaging them into specific geographical regions). $PM_{2.5}$ mass concentrations were the highest in India (n = 18; median: 228 µg $m^{-3}$; Supplementary Fig. 6), while West Midlands had the lowest concentration (n = 21; range: 2–17 µg $m^{-3}$; median: 6 µg $m^{-3}$), followed by Midwest US (n = 241; median: 11.0 µg $m^{-3}$). $PM_{2.5}$ mass concentrations in Chile (n = 85; range: 5–127 µg $m^{-3}$; median: 24 µg $m^{-3}$) were higher than in the Midwest US but much lower than in India. Notably, the range of normalized extrinsic OP and cytotoxicity was higher than the normalized $PM_{2.5}$ mass concentrations in all the regions (Fig. 1).

To further quantify and compare relative variabilities in $PM_{2.5}$ mass vs. OP or cytotoxicity, we calculated coefficients of variation (CoVs) for $PM_{2.5}$ mass, extrinsic OP, and cytotoxicity in different regions, which are shown in Fig. 2. The CoVs of intrinsic endpoints in these regions are shown in Supplementary Fig. 7. Since CoV is more sensitive to the outliers and can be inflated in the cases when arithmetic average of the data approaches zero, we also quantified the

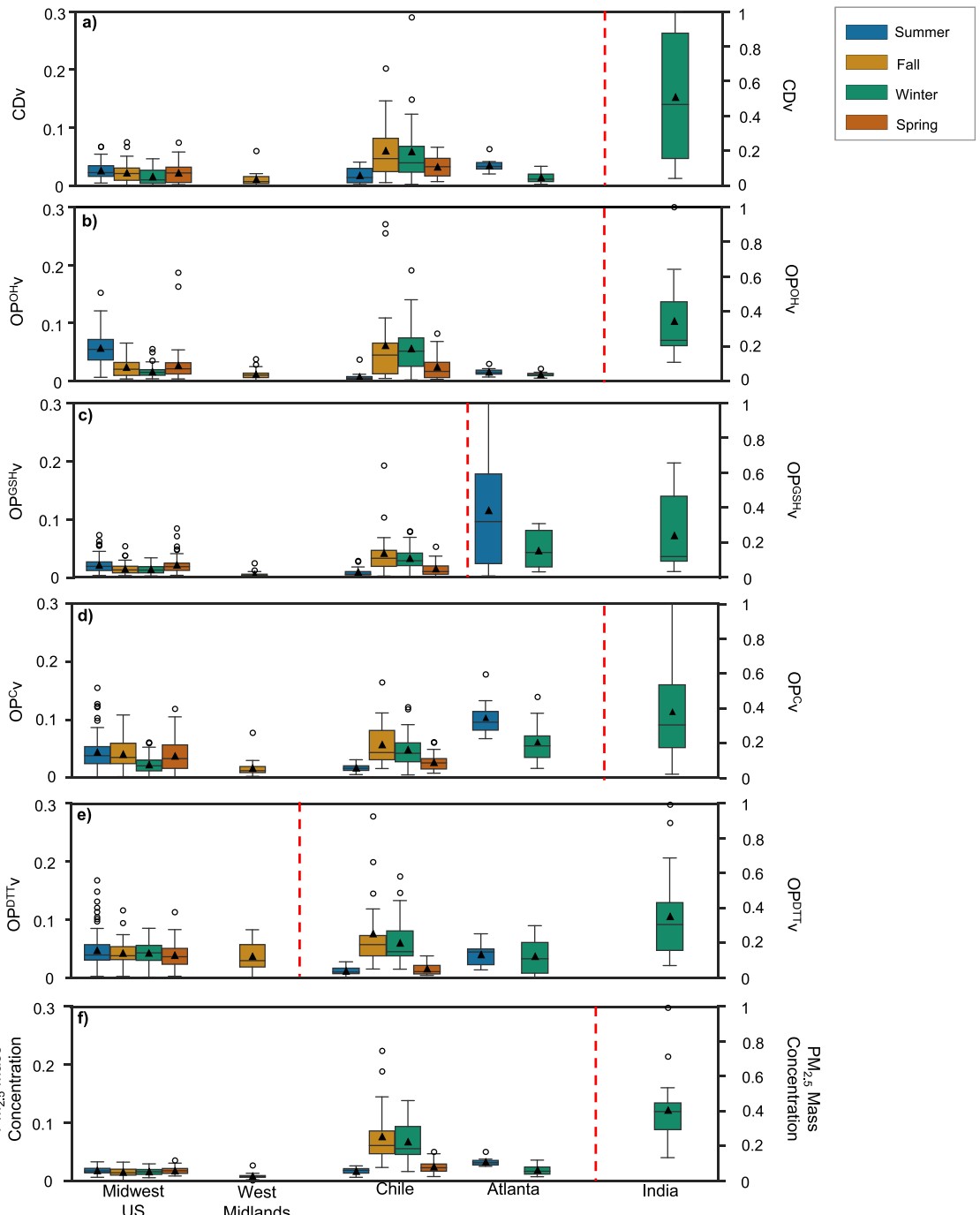

**Fig. 1 | Seasonal distribution of normalized extrinsic cytotoxicity (CDv), oxi-dative potential (OPv), and PM$_{2.5}$ mass concentration in different geographical regions.** The six panels are (**a**) CDv; (**b**) extrinsic acellular OP measured using hydroxyl radical generation rate (OP$^{OH}$v); (**c**) extrinsic acellular OP measured using glutathione depletion rate (OP$^{GSH}$v); (**d**) extrinsic cellular OP (OP$^{C}$v); (**e**) extrinsic acellular OP measured using dithiothreitol depletion rate (OP$^{DTT}$v); and (**f**) PM$_{2.5}$ mass. In all the panels, the bars on the left side of the red dotted line must be read against the primary Y-axis (left side), and the bars on the right side of the red dotted line must be read against the secondary Y-axis (right side). In the Midwest US, samples were collected in summer [n = 69 for all endpoints except for OP$^{C}$v (n = 66) and CDv (n = 65)], fall [n = 56 for all endpoints except for OP$^{C}$v (n = 52), CDv (n = 50) and OP$^{OH}$v (n = 54)], winter [n = 57 for all endpoints except OP$^{C}$v (n = 56) and CDv (n = 54)] and spring

[n = 59 for all endpoints except CDv (n = 54) and OP$^{OH}$v (n = 57)]. Similarly, in Chile, samples were collected in summer [n = 15 for all endpoints except OP$^{C}$v and CDv (n = 13)], fall [n = 20 for all endpoints), winter [n = 30 for all end-points except OP$^{OH}$v (n = 29)] and spring [n = 20 for all endpoints except OP$^{DTT}$v (n = 19)]. In West Midlands, samples were collected only in fall [n = 21 for all endpoints except OP$^{DTT}$v (n = 20)] and in India, samples were collected only during winter [n = 18 for all endpoints except OP$^{DTT}$v (n = 17)]. In Atlanta, samples were collected in summer and winter (n = 10 each, for all endpoints). The box contains the 25–75th percentile of the measurements, the center line of the box denotes the median, and the whiskers denote 1.5 times the interquartile range of the respective endpoints. The black triangle represents the mean. Figure made using Seaborn[93]. Source data are provided as a Source Data file.

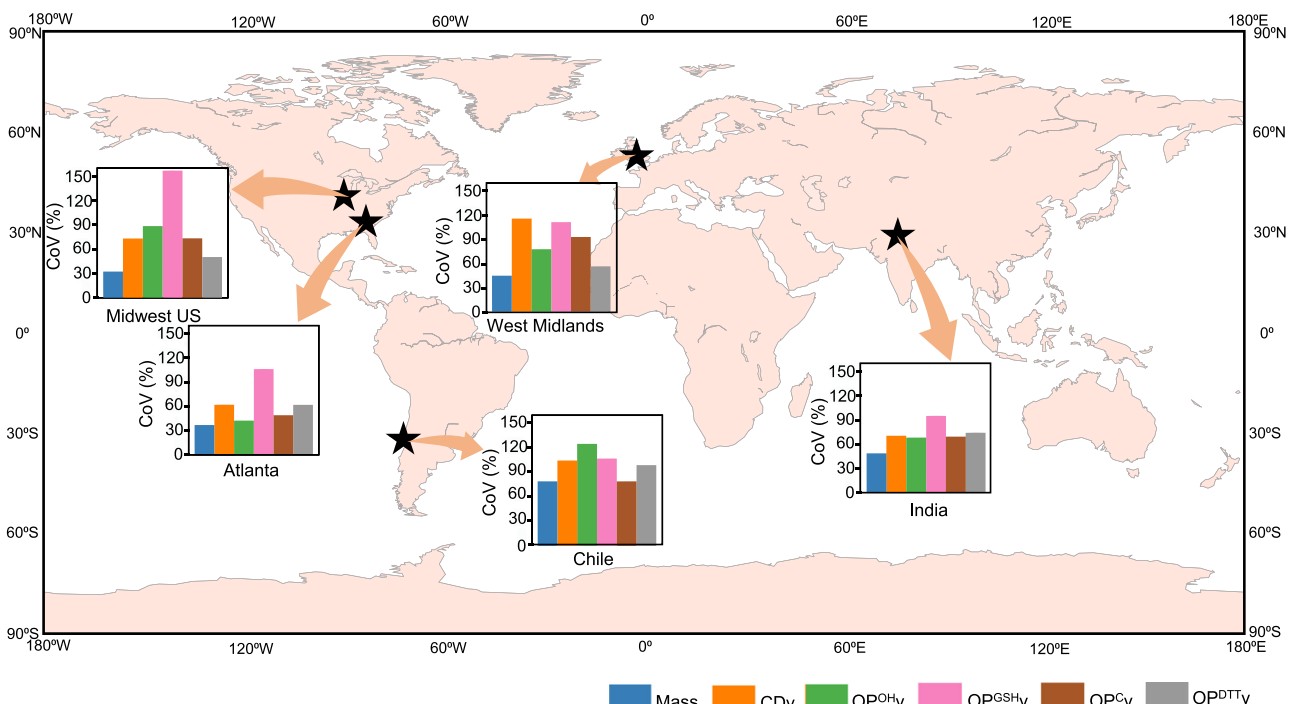

**Fig. 2 | Coefficient of Variation [CoV (%)] for PM$_{2.5}$ mass concentration and various extrinsic PM$_{2.5}$ cytotoxicity (CDv) and oxidative potential (OPv) endpoints.** CoVs are shown for extrinsic endpoints i.e., mass, acellular and cellular OP, and CD for various geographical regions, i.e., Midwest US [n = 241 for mass and extrinsic acellular OP measured using glutathione depletion rate (OP$^{GSH}$v), n = 237 for extrinsic acellular OP measured using hydroxyl radical generation rate (OP$^{OH}$v), n = 223 for CDv and n = 233 for extrinsic cellular OP (OP$^C$v)], Atlanta (n = 20), Chile [n = 85 for mass and OP$^{GSH}$v, n = 84 for extrinsic acellular OP measured using dithiothreitol depletion rate (OP$^{DTT}$v) and OP$^{OH}$v and n = 83 for CDv and OP$^C$v], West Midlands[n = 21 for all endpoints except OP$^{DTT}$v (n = 20)] and India [n = 18 for all endpoints except OP$^{DTT}$v (n = 17)]. Figure made using Matplotlib[94]. Source data are provided as a Source Data file.

variations in different endpoints using two more metrics: robust coefficient of variation based on interquartile range (RCV$_Q$) calculated by Eq. (2)[41]:

$$RCV_Q = 0.75 \times \frac{\text{interquartile range}}{\text{median}} \times 100 \qquad (2)$$

and robust coefficient of variation based on median (RCV$_M$) calculated by Eq. (3)[41]:

$$RCV_M = 1.483 \times \frac{\text{median absolute deviation}}{\text{median}} \times 100 \qquad (3)$$

The RCV$_Q$ and RCV$_M$ values for different extrinsic endpoints (i.e., PM$_{2.5}$ mass, OPv, and CDv) at various sites are given in Supplementary Fig. 8. Interestingly, despite the longest sampling span in the Midwest US (one year), the variation in PM$_{2.5}$ mass concentrations (CoV = 32%; Fig. 2) was lower ($p < 0.05$; see Supplementary Table 3 in SI for statistical significance of the differences in CoVs observed between different sites and endpoints) than in most of the regions except Atlanta. The variation in PM$_{2.5}$ mass concentrations in India was significantly larger (CoV = 49%), compared to the Midwest US ($p = 0.01$) despite only 18 samples collected from India. CoVs of PM$_{2.5}$ mass in Chile (CoV = 77%) were higher than in the Midwest US ($p < 0.001$) and Atlanta ($p = 0.02$). In general, the variations in PM$_{2.5}$ mass were significantly lower than that in OPv and CDv at most sites with few exceptions (see Supplementary Table 3), which was supported by all three metrics used to assess variability, i.e., CoV, RCV$_Q$, and RCV$_M$.

We also calculated the CoV, RCV$_Q$, and RCV$_M$ for the chemical components measured in our study (Supplementary Fig. 8 in SI). A

simple correlation analysis conducted between OP (or cytotoxicity) vs. measured chemical components (Supplementary Table 5 in SI) showed that different OP and cytotoxicity endpoints were associated with different chemical species in different regions. In general, OP$^{OH}$v showed a strong correlation with Fe, Cu, and WSOC ($r > 0.5$), OP$^{DTT}$v was strongly correlated with Fe, Mn, and Cu ($r > 0.6$), and OP$^C$v was associated with Co, Mn, Fe, and Cu ($r > 0.5$). OP$^{GSH}$v showed a moderate correlation with Cu, Al, and K, while CDv showed a strong correlation only with Fe and WSOC ($r > 0.5$). Note, WSOC is a bulk species containing a variety of organic compounds, such as polycyclic aromatic hydrocarbons (PAHs), quinones, carboxylic acids, aldehydes, and amides[42], and measuring the composition of organic aerosols at such a chemically resolved scale is beyond the scope of our current study. Nevertheless, the measured chemical species showed higher variations than PM$_{2.5}$ mass at most sites (see Supplementary Fig. 8). For example, the CoVs for chemical species such as Fe, Mn, and WSOC were 2 times, and CoVs of Cu and Ni were 4 and 7 times greater than the CoVs of PM$_{2.5}$ mass concentrations at all the sites in the Midwest US. In India, although the variability of PM$_{2.5}$ mass was similar to that of WSOC and Fe, other redox-active metallic species such as Cu, Mn, Ni, As, and Pb showed much more (2 times that of PM$_{2.5}$ mass) variability. Similarly, in Chile, although the variability of WSOC was lower than that of PM$_{2.5}$ mass, several redox-active metallic species (Fe, Cu, Mn, Ni, and Cr) varied significantly (2 times than that of PM$_{2.5}$ mass). In West Midlands, the variation in PM$_{2.5}$ mass was similar to the variation in the concentrations of Mn and WSOC; however, Fe and Ni showed higher CoVs (2 times than that of PM$_{2.5}$ mass). Both RCV$_Q$ and RCV$_M$ showed a similar trend as CoV, i.e., higher values for chemical components than PM$_{2.5}$ mass. Thus, the higher variability in OP and cytotoxicity is attributed to larger spatiotemporal variations in redox-active chemical components than PM$_{2.5}$ mass.

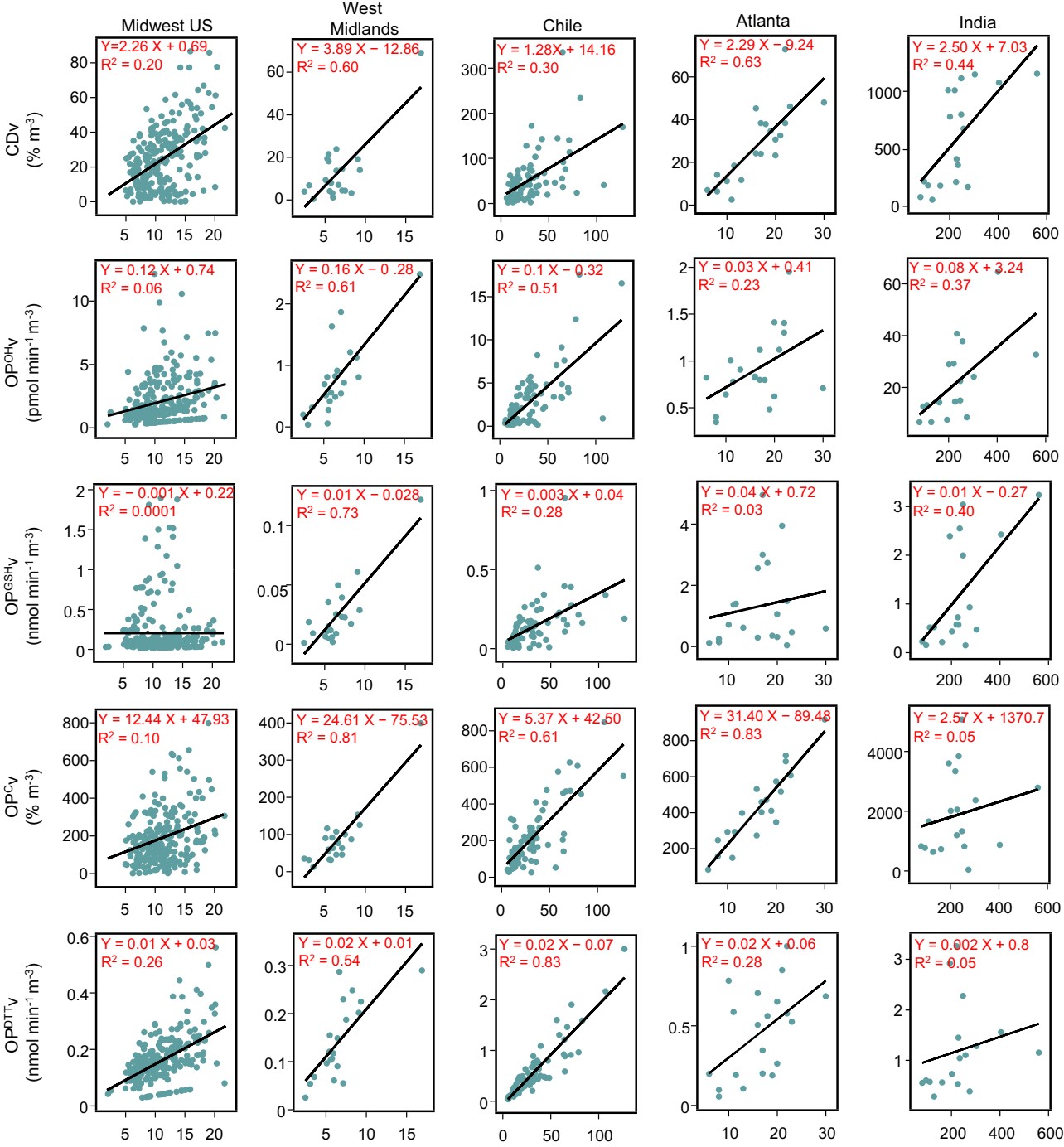

**Fig. 3 | Relationship between extrinsic oxidative potential (OPv) [and extrinsic cytotoxicity (CDv)] vs. PM₂.₅ mass concentrations at different geographical regions.** Simple linear regression for OPv and CDv vs. PM₂.₅ mass concentrations are shown for five geographical regions, i.e., Midwest US [n = 241 for extrinsic acellular OP measured using glutathione depletion rate (OP$^{GSH}$v), n = 237 for extrinsic acellular OP measured using hydroxyl radical generation rate (OP$^{OH}$v),

n = 223 for CDv, and n = 233 for extrinsic cellular OP (OP$^{C}$v)], Atlanta (n = 20), Chile [n = 85 for OP$^{GSH}$v, n = 84 for extrinsic acellular OP measured using dithiothreitol depletion rate (OP$^{DTT}$v) and OP$^{OH}$v, and n = 83 for CDv and OP$^{C}$v], West Midlands [n = 21 for all endpoints except OP$^{DTT}$v (n = 20)] and India [n = 18 for all endpoints except OP$^{DTT}$v (n = 17)]. Figure made using Plotly[95]. Source data are provided as a Source Data file.

## Relationship between PM₂.₅ mass and OP/cytotoxicity

Figure 3 shows a simple linear regression between extrinsic OP (and CD) vs. PM₂.₅ mass concentrations (µg m$^{-3}$). A heatmap of the correlation coefficients (Pearson's $r$) among all the five different endpoints is shown in SI (see Supplementary Fig. 9 and the related discussion in Supplementary Discussion 1). Interestingly, the relationships between

extrinsic OP (and CD) vs. PM₂.₅ mass varied from site to site. For example, there was a strong correlation ($R^2 > 0.5$) between PM₂.₅ mass and almost all the OPv endpoints and CDv for the samples collected in West Midlands. Similarly, for the samples collected in Chile, PM₂.₅ mass showed a strong correlation with almost all the endpoints except for OP$^{GSH}$v and CDv with which it had a moderate correlation ($R^2 < 0.3$).

However, none of these endpoints had a strong correlation ($R^2 < 0.3$) with $PM_{2.5}$ mass for the samples collected from Midwest US, whereas only CDv ($R^2 = 0.44$), $OP^{GSH}v$ ($R^2 = 0.40$), and $OP^{OH}v$ ($R^2 = 0.37$) were moderately correlated with $PM_{2.5}$ mass for the samples collected from India. For samples collected in Atlanta, $PM_{2.5}$ mass displayed a strong correlation only with $OP^Cv$ ($R^2 = 0.83$) and CDv ($R^2 = 0.63$) and a moderate correlation with $OP^{DTT}v$ ($R^2 = 0.28$). More importantly, even for the sites showing strong correlations, the slopes (OPv or CDv vs. $PM_{2.5}$ mass) varied substantially, indicating differential levels of intrinsic toxicities among these sites. For example, although $OP^Cv$ is strongly correlated with $PM_{2.5}$ mass at Atlanta, Chile, and West Midlands, the slope for West Midlands is nearly five times higher and the slope for Atlanta is six times higher than that of Chile. Similarly, although $OP^{OH}v$ is strongly correlated with $PM_{2.5}$ mass for both Chile and West Midlands, the slope for West Midlands is nearly two times higher than that for Chile.

## Importance of intrinsic OP or cytotoxicity of $PM_{2.5}$

Figure 4 plots the entire dataset collected from all the sites, to investigate the relationship between $PM_{2.5}$ mass, OP, and cytotoxicity of $PM_{2.5}$ across the spatial scale encompassed in our study. Figure 4a shows the linear regression analysis of extrinsic OP and cytotoxicity vs. $PM_{2.5}$ mass for the regional averages, while Fig. 4b shows the same regression plot for the entire dataset using both linear and non-linear regression curves. We used logistic regression to model the non-linear relationship between OPv (and CDv) vs. $PM_{2.5}$ mass. Logistic regression has been extensively used in several epidemiological studies to explain the relationships between $PM_{2.5}$ mass and health effects, such as the relationship between exposure to $PM_{2.5}$ and under-5 mortality in China[43], all-cause mortality in the US[44], asthma morbidity in rural USA[45], acute myocardial infarction in USA[46], and elevated platelet counts in Taiwanese adults[47]. Details about various parameters and the software package used to fit the logistic regression curve are given in SI (see Supplementary Method 6). A key message from Fig. 4a is that overall, there is a decent correlation between OPv and $PM_{2.5}$ mass, demonstrating that mass plays a very important role in determining the overall toxicity and possibly the health effects of the aerosols. For example, West Midlands, UK, which had the lowest average $PM_{2.5}$ mass concentrations, had the lowest average $OP^{DTT}v$, $OP^{GSH}v$, $OP^Cv$, and CDv. Similarly, India, which had the highest average $PM_{2.5}$ mass concentrations, had the highest average $OP^{DTT}v$, $OP^{OH}v$, $OP^Cv$, and CDv. This justifies a strong and consistent relationship between $PM_{2.5}$ mass and mortality/morbidity observed worldwide in epidemiological studies[48–52]. However, one must exercise caution in generalizing these results, because there is a significant scatter in these plots, which are somewhat hidden because of India results being substantially higher than the rest of the sites [note a significant drop in $R^2$ and an increase in root mean squared error (RMSE) after removing India results in most cases]. This scatter can be explained by differences in intrinsic toxicities of $PM_{2.5}$ at these sites. The intrinsic $PM_{2.5}$ toxicity in India was significantly lower than most sites for almost all the endpoints. $PM_{2.5}$ samples collected from Midwest USA during fall and summer seasons had significantly higher intrinsic cytotoxicity and OP ($p < 0.05$) compared to the $PM_{2.5}$ collected from West Midlands, India, and Chile, irrespective of the endpoints. Similarly, $PM_{2.5}$ samples collected from Atlanta had significantly higher intrinsic OP (for three endpoints: $OP^{DTT}m$, $OP^{GSH}m$, and $OP^Cm$; $p < 0.05$) compared to the $PM_{2.5}$ samples collected from West Midlands, India, Chile (spring and summer) and Midwest US sites. In fact, Atlanta had the highest average $OP^{DTT}m$ (0.03 $min^{-1} \mu g^{-1}$), $OP^{GSH}m$ (0.08 $nmol\ min^{-1} \mu g^{-1}$), and $OP^Cm$ (25% $\mu g^{-1}$) among all the sites (see Supplementary Fig. 6).

This difference in intrinsic toxicity of the $PM_{2.5}$ actually results in a non-proportional relationship between $PM_{2.5}$ mass and extrinsic OP and cytotoxicity, i.e., the ratios of $PM_{2.5}$ mass measured at two sites were much higher than the ratios of extrinsic OP at those sites. For

example, the average $PM_{2.5}$ mass concentration in the Midwest US was 20 times lower than that in India; however, the average $OP^{GSH}v$, $OP^{OH}v$, $OP^{DTT}v$, and $OP^Cv$ were only 5, 10, 7, and 10 times lower, respectively (Supplementary Fig. 6). Similarly, although the average $PM_{2.5}$ mass concentration during the fall season in Chile was 4 times higher than that in the Midwest US, average $OP^{OH}v$, and $OP^Cv$ were only 3 and 2 times higher, respectively. In some cases, the ratios of $PM_{2.5}$ mass concentrations between the sites were lower than the respective ratios of extrinsic OP. For example, although the average $PM_{2.5}$ mass concentration in Atlanta was 3 times higher than that in West Midlands, the $OP^{GSH}v$ and $OP^Cv$ were 53 and 5 times higher than that in West Midlands, respectively. On the contrary, $PM_{2.5}$ mass and OPv were in fact inversely related for some site pairs. For example, although the average $PM_{2.5}$ mass concentrations during the winter season were significantly higher in Chile as compared to Atlanta (3 times), the $OP^{GSH}v$ in Atlanta was 5 times higher than the average $OP^{GSH}v$ in Chile.

This non-proportionality between OPv and $PM_{2.5}$ mass becomes more apparent when we plotted the entire dataset from all the sites instead of the regional averages (Fig. 4b). Interestingly, the OPv (and CDv) vs. $PM_{2.5}$ mass curves exhibit a non-linear trend, with a steep slope at lower $PM_{2.5}$ mass (<50 $\mu g\ m^{-3}$) and gradual flattening at higher mass concentrations (>300 $\mu g\ m^{-3}$). Note, there is a significant increase in $R^2$ and a decrease in RMSE when replacing a linear curve with a non-linear fitting curve for almost all of the endpoints (except $OP^{GSH}v$). These results, which are strikingly consistent with newer epidemiological studies demonstrating a supralinear relationship between $PM_{2.5}$ mass and mortality observed at lower $PM_{2.5}$ concentrations[53] and the flattening of the CR curve at higher $PM_{2.5}$ concentrations[54], provide an important mechanistic basis for the non-linear relationship between $PM_{2.5}$ mass and health effects.

We hypothesize that differences in the slopes of the OPv and CDv vs. $PM_{2.5}$ mass at different sites (Fig. 3) and the resultant non-linearity in the OPv trend in the entire dataset (Fig. 4b) is largely caused by substantial differences in the $PM_{2.5}$ chemical composition among different regions. To further test this hypothesis, we chose the dataset obtained from Chile because that is the only region among our study sites where $PM_{2.5}$ mass ranged most widely from 5 to 127 $\mu g\ m^{-3}$, with minimal variation in the chemical composition. As can be seen from Supplementary Fig. 8, unlike other sites, the CoVs for various relevant chemical components known to be redox-active (e.g., Cu[55–57], Mn[31,56] and WSOC[18,58]) are in the similar range as for $PM_{2.5}$ mass at both sites in Chile (Chillan and Santiago). Interestingly, despite such a large variation in the $PM_{2.5}$ mass concentration, the relationship between OPv (or CDv) vs. $PM_{2.5}$ mass is largely linear for most endpoints, with an $R^2$ ranging from 0.28 to 0.83, which doesn't improve further by attempting a flattening curve at higher $PM_{2.5}$ concentration ranges (see Supplementary Fig. 10). It suggests that the use of a fixed CR curve based on $PM_{2.5}$ mass for a given region is reasonable, so far there are no substantial spatial or temporal changes in the chemical composition.

Based on a presumption that OP is closely related to the health effects (as suggested by the studies discussed earlier showing a stronger association of OP with distinct toxicological/clinical endpoints than $PM_{2.5}$ mass), our results imply that the relationship between $PM_{2.5}$ and health effects is not solely driven by $PM_{2.5}$ mass and the role of chemical composition which drives its intrinsic toxicity cannot be ignored. Our results emphasize the need for developing region-specific CR curves, rather than using a generalized curve globally. This conclusion is supported by various recent epidemiological studies which have found that the hazard ratio estimates from cohort studies in China were much different from those of Integrated CR function estimates[59,60], that risk of all-cause mortality can vary between different regions within a country[60], and that there were clear urban-rural disparities in the association of mortality and $PM_{2.5}$ mass[61–63]. Thus, using globally generalized linear CR curves

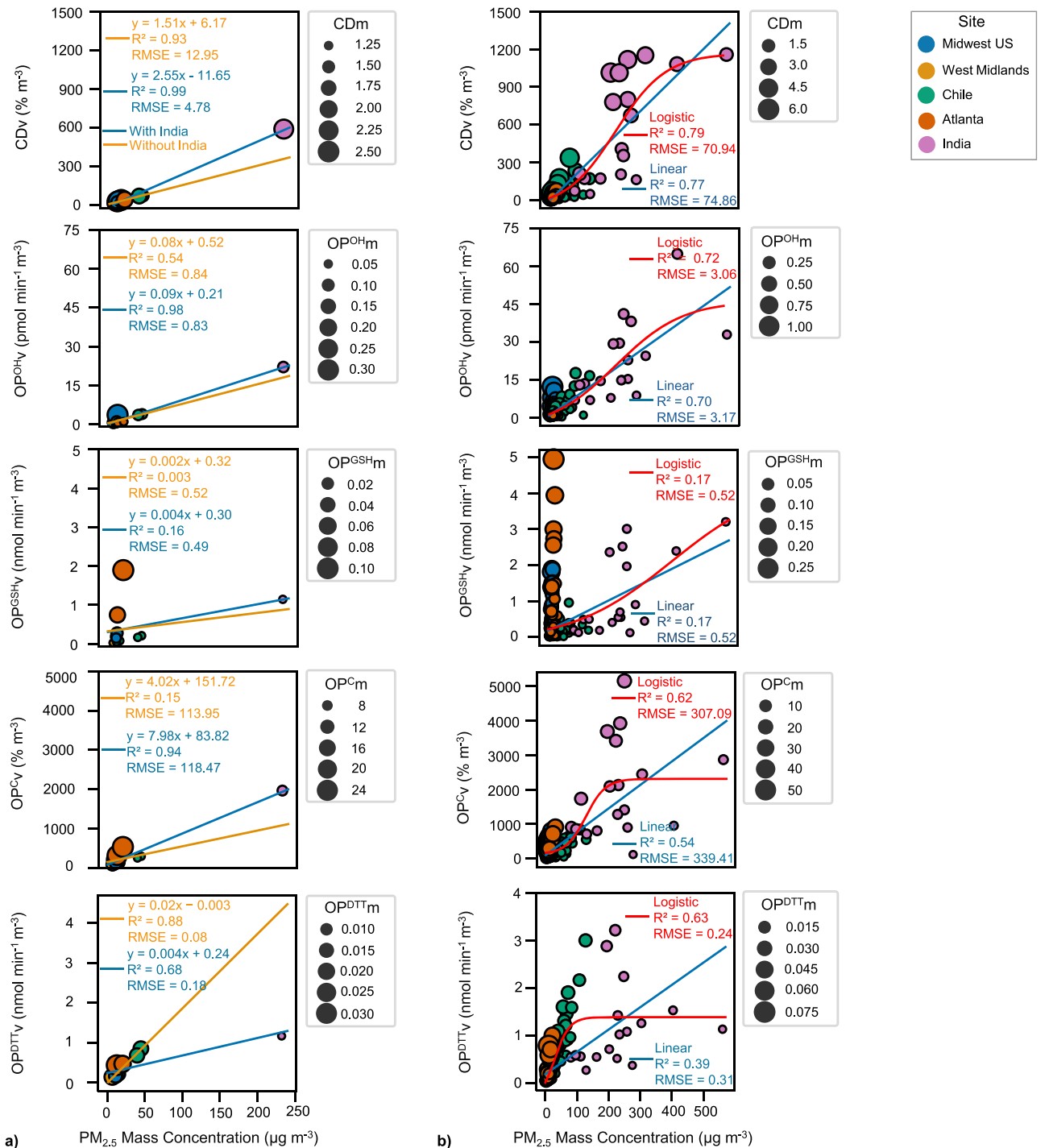

**Fig. 4 | Relationship between extrinsic oxidative potential (OPv) [and extrinsic cytotoxicity (CDv)] vs. PM$_{2.5}$ mass concentrations based on the entire dataset plotted together.** Here, the intrinsic OP and cytotoxicity are represented by the size of the bubble. Curves are fitted for (**a**) the seasonally averaged data for all five regions considered in this study (n = 12); and (**b**) the entire dataset for all sites: extrinsic acellular OP measured using dithiothreitol depletion rate (OP$^{DTT}$v) (n = 382), extrinsic cellular OP (OP$^C$v) (n = 375), extrinsic acellular OP measured using glutathione depletion rate (OP$^{GSH}$v) (n = 385), extrinsic acellular OP measured using hydroxyl radical generation rate (OP$^{OH}$v) (n = 380) and CDv (n = 365). In (**a**), the blue line represents the linear curve fitted for all five regions and the orange line represents the linear curve after excluding India. In (**b**), the blue line represents a linear curve, and the red line represents the fitted logistic curve. RMSE represents root mean squared error. Figure made using Seaborn[93]. Source data are provided as a Source Data file.

to predict the health effects of PM$_{2.5}$ in high PM$_{2.5}$ concentration regions such as China and India might cause unrealistic estimations of morbidity and mortality[64]. Consequently, the policy measures focused on reducing PM$_{2.5}$ concentrations alone based on globally generalized linear CR curves would not yield equivalent reductions in the health effects[65].

## Limitations and implications

Although substantial efforts were resourced in our study to coordinate PM$_{2.5}$ sampling in various parts of the world, it had some limitations which should be carefully considered before the general implication of our results. First, the collection and transport of filters from such an extensive spatial scale leads to unavoidable artifacts related to offline

filter collection and the variable periods of sample storage, which could result in the loss of short-lived redox-active compounds, e.g., peroxy radicals and peroxide-containing highly oxygenated molecules (HOMs)[66–68], and semi-volatile organic compounds[67]. Thus, the OP and cytotoxicity of the PM$_{2.5}$ samples collected in our study could have been underestimated. Second, our study focused only on water-soluble extracts of PM$_{2.5}$. Water-insoluble species of PM$_{2.5}$ have also been shown to contribute to PM$_{2.5}$ toxicity[33,69,70], and thus water-soluble fraction of the PM$_{2.5}$ used in our study can be considered as the lower limit, as it does not fully substitute for the overall toxicity of PM$_{2.5}$. Note, our choice of water-soluble fraction was driven by the lack of a standardized protocol to measure total OP or cytotoxicity of PM$_{2.5}$ that is equally applicable to both acellular and cellular assays. Although, several solvents (e.g., methanol, dichloromethane, hexane, acetone, and acetonitrile) have been suggested to extract the water-insoluble fraction of PM$_{2.5}$[70–72], the adequacy of these solvents to retain PM$_{2.5}$ chemical composition, which is physiologically relevant for the cellular exposure has not been tested. The choice of our cell line and the endpoints could also limit the implication of cellular toxicity results. Although, A549 is a widely used alveolar epithelial cell line relevant to alveolar exposure to PM$_{2.5}$, its responses cannot be equated to other cell lines such as BEAS-2B, 16-HBE14o, Calu-3 (relevant to broncho tracheal region exposure to PM$_{2.5}$), and pulmonary and cardiovascular cell lines [e.g., THP-1 (macrophages), HEK-293, HMVEC-L and HULEC-5a (human lung microvascular endothelial cells), and H9C2]. Measurement of cellular responses other than cellular OP and cytotoxicity, such as inflammatory cytokines, gene expressions, and specific type of cell death (e.g., necrosis, apoptosis, and autophagy) could provide more valuable insights into the toxicity mechanisms triggered by PM$_{2.5}$. We also acknowledge that although we measured the time-dependent responses of acellular assays, given the laborious protocols of cellular assays, we followed the conventional method based on measuring cellular responses only at 24 h, which could underestimate some of these endpoints.

Although, the overall range of PM$_{2.5}$ mass concentrations obtained from our samples is quite large (2–561 µg m$^{-3}$), it is still mostly dominated by the samples with mass concentrations <50 µg m$^{-3}$ (354 out of 385 samples), with only 19 samples having mass concentrations in the range of 50–200 µg m$^{-3}$. Thus, the curves shown in Fig. 4 could be somewhat biased by the samples with low PM$_{2.5}$ mass concentrations. Future studies should focus on more measurements at medium and high PM$_{2.5}$ concentrations to better constrain the non-proportional relationships between health metrics and PM$_{2.5}$ mass. Finally, we acknowledge that developing localized CR curves for specific regions will require substantial efforts on the part of epidemiological research. Moreover, the variabilities in the PM$_{2.5}$ chemical composition caused by other factors such as varying weather, changing landscape of emissions (e.g., introduction of electric vehicles, etc.), makes it even more complicated to ascertain the boundary conditions (i.e., time and space) for conducting such epidemiological studies. Therefore, in addition to these studies, we suggest that alternative metrics which can better represent the array of health effects associated with PM$_{2.5}$ pollution should be explored. OP could be one of such metrics, but more world-wide studies (such as the current one or even more extensive in terms of space and time) need to be conducted to understand its spatiotemporal distribution and test its health relevance, by integrating them in epidemiological studies.

## Methods
### Sampling site and sampling periods
A total of 385 ambient PM$_{2.5}$ samples were collected from 14 different sites on four different continents. These include five sites in the Midwest US: Champaign, IL (51 samples), Chicago, IL (44 samples), Indianapolis, IN (54 samples), St. Louis, MO (47 samples) and Bondville, IL (45 samples); two sites in Chile: Santiago (50 samples), Chillan

(35 samples); one site in Southeast US: Atlanta (20 samples); two sites in West Midlands, Birmingham (UK): a roadside site [EROS (8 samples)] and an urban background site [BROS (13 samples)], and four sites in India: Ahmedabad (3 samples), Hisar (5 samples), Patiala (5 samples) and Faridabad (5 samples). The sampling dates and respective PM$_{2.5}$ mass concentrations for all the sites are provided in Supplementary Table 1 of SI.

Quartz filters were used for collecting PM$_{2.5}$ at all the sites, except in West Midlands and Chile, where Teflon and glass fiber filters were used, respectively. Previous studies have shown that the measurement of OP is not significantly influenced by the type of filter for commonly used OP assays such as OP$^{DTT}$ [ref. 69 (slope = 1.26 and R$^2$ = 0.88 for quartz vs. Teflon filters) and ref. 73 (slope = 1.05 and R$^2$ = 0.98 for glass fiber vs. Teflon filters)] and OP$^{OH}$ (Shen and Anastasio[74]) as long as a consistent extraction procedure is followed for all filters. The sampling equipment and site-related information for the PM$_{2.5}$ samples collected in the Midwest US have been discussed in detail in our previous publications[17,18,33]. Briefly, PM$_{2.5}$ samples (integrated samples for a continuous sampling duration of 72 h) were collected using a Hi-Vol sampler (flow rate: 1.13 m$^3$ min$^{-1}$) between May 22, 2018, and May 30, 2019. Similarly, PM$_{2.5}$ samples (continuous sampling duration of 24 h) were collected at Jefferson Street site in Atlanta using a Hi-Vol sampler (flow rate: 1.13 m$^3$ min$^{-1}$) between January 26, 2018, and December 26, 2018[55]. This site is located roughly 4 km northwest of downtown Atlanta and is representative of the urban Atlanta region. In India, 10 h (for Patiala) and 24 h (for Ahmedabad, Hisar, and Faridabad) integrated samples were collected using a Hi-Vol sampler (Thermo Scientific, USA, flow rate: 1.13 m$^3$ min$^{-1}$). Samples in Patiala were collected between October 26, 2011, and February 4, 2012, whereas for the rest of the sites, samples were collected during the months of late October and November in 2019 and 2020. The sampling site in Patiala was located on the terrace of the Department of Physics, Punjabi University, Patiala, ~20 m above ground level (AGL)[75]. The sampling site in Hisar was located at the Agrimet Observatory, Chaudhary Charan Singh Haryana Agricultural University (CCSHAU), Hisar. The site was at the ground level and <200 m away from a moderately busy road[76]. Sampling in Faridabad was carried out at Manav Rachna International Institute of Research and Studies [MRIIRS; second floor of the C block building (~7 m AGL)]. The site was close to a busy road with traffic load from heavy-duty trucks[76]. The sampling in Ahmedabad was carried out at the rooftop of a multi-storied building (~50 m AGL) of the Physical Research Laboratory using a Hi-Vol sampler (Thermo Scientific, flow rate: 1.13 m$^3$ min$^{-1}$). Ahmedabad is an urban city (population > 7 million) in the semi-arid region of western India with many industries and thermal power plants in the surroundings[77].

24 h integrated PM$_{2.5}$ samples (from midnight to midnight) in two cities of Chile − Chillan and Santiago, were collected using a Hi-Vol sampler [model CAV-A/mb, MCV SA, Barcelona, Spain (flow rate: 0.5 m$^3$ min$^{-1}$)] between December 12, 2018, and January 14, 2020. The Santiago metropolitan sampling site was situated on the rooftop of the Faculty of Science at the University of Chile. The site is in proximity to both residential and commercial centers, with a significant presence of vehicular traffic. The Chillan sampling site was located on the rooftop of the University of Bío-Bío, which is located within the central business district. 120 h integrated PM$_{2.5}$ samples in Birmingham (UK) were collected using a Hi-Vol sampler [Ecotech HiVol 3000 (flow rate: 0.5 m$^3$ min$^{-1}$)] between August 12, 2019, and November 26, 2019. The sampling site in Birmingham was located at the 'Birmingham Air Quality Supersite', within the Edgbaston campus of the University of Birmingham. Nearby potential anthropogenic emission sources include a suburban rail line (located approximately 90 m northwest of the site) and a suburban road (~125 m east of the site)[78]. After sampling, the filters collected from all the sites were gravimetrically weighed at the respective laboratory facility near the site, except Atlanta where PM$_{2.5}$ mass concentrations were measured using a Tapered Element

Oscillating Microbalance (TEOM). The filters were subsequently stored in hermetically sealed containers at a temperature of −20 °C until shipped to the University of Illinois at Urbana-Champaign (UIUC) in a thermally insulated box containing blue ice to minimize PM$_{2.5}$ loss and stored immediately in a freezer (at −20 °C) upon arrival. All the filters were analyzed for OP, cytotoxicity, and chemical composition analyses within 1 year of storage at UIUC. The details of various instruments used for PM$_{2.5}$ sample collection, and measurement of mass, chemical composition, OP, and cytotoxicity are provided in Supplementary Table 2.

## Extraction of PM$_{2.5}$ filters

PM$_{2.5}$ water-soluble extracts were prepared by immersing a single circular section of 2.5 cm diameter from the PM$_{2.5}$ filters in deionized water (DI; Milli-Q; resistivity = 18.2 MΩ cm$^{-1}$) and sonicating in an ultrasonic water bath (Cole-Parmer, Vernon Hills, IL, USA) for 1 h. The volume of DI was adjusted such that the final concentration of the extract for exposure in the reaction vial (RV) for both cellular and acellular assays was 30 μg mL$^{-1}$ (please see the section "Cellular OP and cytotoxicity measurements" for the justification of this concentration). After sonication, the extracts were passed through a 0.45 μm pore size polytetrafluoroethylene (PTFE) filter to remove any insoluble particles and/or filter fibers. These water-soluble extracts were then analyzed for cellular OP, cytotoxicity, acellular OP, and chemical composition analyses.

## Cell culture

In this study, a lung epithelial cell line, A549 (adenocarcinoma human alveolar basal epithelial cells; ATCC CCL-185), which is used as a model of Type II lung epithelium cells, was used to measure the impact of PM$_{2.5}$ water-soluble extracts on cell viability and cellular OP. A549 is one of the most widely used cell lines in PM$_{2.5}$ toxicological studies[79–81], and is representative of the cells responsible for the diffusion of substances, such as water and electrolytes across alveoli of the lungs. The cells in this region play a crucial role in preventing inflammation[82] and maintaining the normal lung architecture by renewing other types of alveolar cells[83]. Moreover, the alveolar region facilitates the entrainment of particles and their constituents into other regions of the body by crossing the blood-air barrier[84], making A549 a suitable choice for our study. Cells were grown in Ham's F-12K (Kaighn's) culture medium, consisting of 15% heat-inactivated fetal bovine serum, 2 mM L-glutamine, and 1500 mg L$^{-1}$ sodium bicarbonate. Plastic petri dishes with a cell concentration of $1 \times 10^4$ viable cells cm$^{-2}$ were used to culture the cells. The petri dishes were kept in a humidified incubator at 5% CO$_2$ and 37 °C and subcultures were prepared once every week. The medium was renewed 2 times per week.

## Cellular OP and cytotoxicity measurements

The cellular OP measurement protocol was adapted from previous studies which used DCFH-DA to measure intracellular ROS[18,85–87]. In the first step, $1 \times 10^4$ cells suspended in 200 μL of Ham's F-12K culture medium were added to the wells of 96-well plates and incubated for 24 h. After incubation, the culture medium was aspirated, and the cells were washed thrice with PBS. In the second step, PM$_{2.5}$ water extract (78 μL), a working solution of 450 μM DCFH-DA (22 μL; DCFH-DA preparation details are given in the Supplementary Method 1 in SI) and culture medium (100 μL), were added and the cells were incubated in the dark for 24 h. The exposure duration of 24 h was chosen because the working solution of DCFH-DA was found to be stable till at least 25 h in our initial experiments. Details of the experiment showing the time-dependent absolute fluorescence of DCFH-DA are provided in Supplementary Method 2 and Supplementary Fig. 1 in SI. After incubation, the cell culture media (also containing dead cells floating in the media, if any) from each well was aspirated and transferred to separate 2 mL vials. Then, 30 μL of 0.25% trypsin (with 2.21 mM EDTA) was

added to each well to detach the viable cells (attached to the bottom of the plate) and incubated for 4 min, following which the constituents of the wells were aspirated and transferred to their respective 2 mL vials. Finally, an aliquot of 60 μL was taken from each vial and diluted 100 times, before measuring its fluorescence intensity at 488 nm excitation and 532 nm emission wavelengths using a bench-top spectro-fluorometer (RF-5301 pc, Shimadzu Co., Japan). The measured fluorescence is proportional to the amount of ROS generated as a result of PM$_{2.5}$ exposure to the cells. The fluorescence intensity of negative control (a blank filter extracted in DI and exposed to the cells in the same way as the PM$_{2.5}$ sample) was subtracted from the fluorescence intensity of each sample and OP results are reported as the percentage increase in fluorescence relative to the negative control.

For calculation of intrinsic (i.e., normalized by PM$_{2.5}$ mass) and extrinsic (i.e., normalized by volume of air) cellular OP, it is necessary to choose an appropriate PM$_{2.5}$ extract concentration such that it lies in the linear range of the dose-response curve. To determine that, we randomly selected eight PM$_{2.5}$ samples and measured their cellular OP for different concentrations of PM$_{2.5}$ water extracts (5, 10, 20, 30, 50, 100, 200, 300 μg of PM$_{2.5}$ mL$^{-1}$). As shown in Supplementary Fig. 2, the curve is roughly linear for almost all of the PM$_{2.5}$ samples within the concentration range of 20 to 300 μg mL$^{-1}$, justifying the choice of PM$_{2.5}$ extract concentration (30 μg mL$^{-1}$) used in this study for cellular OP measurements. The exact procedure used for intrinsic and extrinsic cellular OP calculations is given in SI (Supplementary Method 3).

Cell viability was measured using crystal violet assay as described in refs. 22,88 with some modifications. In the first step, $1 \times 10^4$ cells suspended in 200 μL of Ham's F-12K culture medium were added to the wells of 96-well plates and incubated for 24 h. After incubation, the culture medium was aspirated, and the cells were washed with PBS. In the second step, PM$_{2.5}$ extract (78 μL) and culture medium (122 μL) were added. All the plates were then sealed using a sterile aluminum film (AlumaSeal®) to prevent cross-contamination and loss of liquid due to evaporation. Six technical replicates (i.e., six consecutive wells in one column of the 96-well plate) were used for each sample and one column (of six consecutive wells) of each microplate served as the negative control comprising solely of culture medium and field blank extracts along with the cells. Two columns (of six consecutive wells) in each microplate comprising solely of culture medium alone without any cells served as the background controls. Two separate experiments were performed on the cells from different subcultures to represent two biological replicates. After 24 h, the culture medium containing PM$_{2.5}$ extracts was removed from the well of the 96-well plate and the cells were fixed by adding 50 μL of methanol. The cells were then stored in the dark for 10 min. After that, the methanol was removed, and the plate was thoroughly tapped to ensure there was no methanol remaining in the wells. The cells were then stained with 1% crystal violet solution in 50% methanol for another 10 min. The plate was then washed thoroughly in a water bath to remove excess crystal violet dye, tapped dry and 100 μL of 75% DMSO (v/v) and 25% (v/v) methanol was added to the wells. The plates were incubated in the dark for 10 min before being analyzed for the absorbance measurement at 595 nm using a Bio Tek Epoch2 microplate reader (Agilent, CA). The absorbance of each well was recorded and corrected for background. The average absorbance of the six wells containing negative control was defined as 100% viability, and viability in the wells containing PM$_{2.5}$ extract was calculated based on their absorbance relative to the negative control. Cytotoxicity or Cell death (CD) was then derived from cell viability using the formula: CD = 100% − cell viability. The detailed procedure to calculate intrinsic and extrinsic CD is given in SI (Supplementary Method 4). Both cellular OP and cytotoxicity were measured on the same day and in the cells obtained from the same culture plates.

Similar to cellular OP, we also ensured that the PM$_{2.5}$ extract concentration chosen to measure cytotoxicity lies in the linear range of

the dose-response curve. We plotted the cell viability vs. $PM_{2.5}$ extract concentration curves for different extract concentrations (5, 10, 20, 30, 50, 75, 100, 200, 300 μg of $PM_{2.5}$ mL$^{-1}$) for the same $PM_{2.5}$ samples used to evaluate extract concentration-cellular OP relationship. As shown in Supplementary Fig. 3, the curve is roughly linear for all $PM_{2.5}$ samples ($r > 0.8$) for concentrations between 25 and 300 μg mL$^{-1}$, which justifies the concentration (30 μg mL$^{-1}$) chosen for our cytotoxicity experiments.

### Acellular OP measurements

We measured the following three OP endpoints – GSH depletion rate, DTT consumption rate, and OH• (OH-SLF) generation rate in a surrogate lung fluid (SLF). SLF was prepared by mixing four different antioxidants to achieve final concentrations of AA, GSH, Uric Acid (UA), and Citric Acid (CA) as 200 μM, 100 μM, 100 μM, and 300 μM, respectively[89]. Similar to cellular OP and cytotoxicity measurements, we used a fixed concentration of 30 μg mL$^{-1}$ of $PM_{2.5}$ extract in the RV for all acellular OP measurements to avoid non-linear dose-response effects caused by certain $PM_{2.5}$ components such as Cu and Mn[90]. Acellular OP measurements were conducted using a semi-automated multi-endpoint ROS-activity analyzer (SAMERA) developed in our lab[17]. The design and operating procedure for SAMERA are described in ref. [17]. DTT consumption rate was measured using the 5,5'-dithiobis-(2-nitrobenzoic acid) (DTNB) method[57]. Briefly, DTNB and a small aliquot from the RV containing a mixture of $PM_{2.5}$ extract and DTT were added to a measurement vial (MV). The DTNB reacts with residual DTT to form a yellow-colored compound called 2-nitro-5-thiobenzoic acid (TNB). TNB was then diluted using DI and passed through a liquid waveguide capillary cell (LWCC-3100; World Precision Instruments, Inc., Sarasota, FL, USA), where the absorbance at 412 nm and 600 nm (background) was measured by the spectrophotometer (Ocean Optics; Dunedin, FL, US). This process was repeated at time intervals of 5, 17, 29, 41, and 53 min to obtain the $PM_{2.5}$-catalyzed DTT decay rate. GSH depletion rate was measured using the o-phthaldialdehyde (OPA) method[17]. In this method, a small aliquot from the RV containing $PM_{2.5}$ extract and SLF was withdrawn at time intervals of 5, 24, 43, 62, and 81 min, and transferred to a MV along with OPA. GSH reacts with OPA to form a fluorescent product called GS-OPA, and the fluorescence was measured at an emission wavelength of 427 nm (excitation wavelength = 310 nm) to estimate the residual GSH concentration. Finally, OH• generation rate was measured using 2-OHTA method[17]. In this method, disodium terephthalate (TPT) is added to the RV containing SLF and $PM_{2.5}$ extract to capture the OH• generated during the reaction of $PM_{2.5}$ with the antioxidants contained in SLF. The reaction between TPT and OH• produces a fluorescent product: 2-OHTA, which was withdrawn from the RV at time intervals of 10, 29, 48, 67, and 86 min, and diluted with DI in a MV. The diluted 2-OHTA was then passed through the flow cell of the spectrofluorometer (Fluoromax-4, Horiba Scientific, Edison, NJ, USA) to measure its fluorescence (excitation: 310 nm; emission: 427 nm). The instrument was calibrated with known standards of 2-OHTA (0-200 nM) and a yield factor of 0.35 (formation of 2-OHTA from OH•) was applied to determine the concentration of OH•. The slopes of the DTT, GSH, and OH• concentration vs. time curves were then used to determine the consumption (in case of DTT and GSH) and generation (in case of OH•) rates (μM min$^{-1}$ and nM min$^{-1}$, respectively) in various OP assays. Detailed information about the positive controls used for these OP endpoints is given in the SI (see Supplementary Fig. 4).

### Chemical composition analyses

We analyzed the water-soluble $PM_{2.5}$ extracts for water-soluble organic carbon (WSOC) using total organic carbon analyzer (TOC analyzer; TOC-VCPH, Shimadzu Co., Japan) and the concentrations of various elemental species (Li, Al, K, V, Cd, Co, Cr, Ni, As, Rb, Sr, Ba, Pb, Zn, Cu, Fe, Ga, and Mn) using inductively coupled plasma mass spectrometer

(ICP-MS; NexION 300, Perkin Elmer, Waltham, MA)[18] as described in Supplementary Method 7.

### Statistical analyses

The normality of the distribution of OP and cytotoxicity measurements was tested using the Shapiro-Wilk test which showed statistically insignificant results ($p > 0.05$) indicating that the data was normally distributed. Simple linear and logistic regression curves between $PM_{2.5}$ mass and OP (or CD) were fitted using the optimization package from an open-source Python library called SciPy and coefficient of determination ($R^2$) and RMSE were used to evaluate the performance of linear and logistic regressions. A two-tailed t-test was conducted to determine the statistical significance of the differences between $PM_{2.5}$ OP (or CD) in various regions. Statistical significance of the differences in CoVs observed between different sites and OP (or cytotoxicity) endpoints was determined by the asymptotic test for the equality of CoVs as proposed by Feltz and Miller[91] using the R package cvequality (Version 0.1.3)[92]. Uncertainties in all measurements were estimated by propagating the uncertainties in various instruments and methodologies used for $PM_{2.5}$ sampling, sample extraction, OP and chemical characterization, and cytotoxicity measurement (see Supplementary Table 2 in SI).

## Data availability

Source data are provided with this paper. https://doi.org/10.6084/m9.figshare.25538341.

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

## Acknowledgements

The authors thank the National Science Foundation (grant no. CBET-1847237 and CBET-2012149) for financial support. We thank Sandra McMasters, the director of the Cell Media Facility at UIUC, for providing us with cell culture media. S.N.T. acknowledges financial support received under the Centre of Excellence Advanced Technologies for Monitoring Air-quality iNdicators (ATMAN) approved by the PSA office, Government of India, and supported by a group of philanthropic funders, including the Bloomberg Philanthropies, the Open Philanthropy, and the Clean Air Fund. S.N.T. also acknowledges support under J. C. Bose Fellowship of Science and Engineering Research Board, Department of Science and Technology. M.A.L.G. acknowledges the partial financial support provided by ANID FONDECYT Regular (grant no. 1241485).

## Author contributions

S.S. and V.V conceptualized research; S.S., H.Y., Z.D., and P.S.G.S. performed research and data analysis; A.S., F.D.P., N.R., S.N.T., R.W., H.Y., Y.W., J.V.P., and M.A.L.G. provided expertise and PM$_{2.5}$ samples; S.S. and V.V. wrote the paper with input from A.S., F.D.P., N.R., S.N.T., R.W., and M.A.L.G., J.V.P., H.Y, and Y.W.

## Competing interests

The authors declare no competing interests.
