## [Peer Review File · Nature Communications]

Inter-continental variability in the relationship of oxidative potential and cytotoxicity with PM2.5 massReviewer #1 (Remarks to the Author):

In this manuscript, the authors assess the connection between particulate matter and oxidative potential by employing 385 PM_{2.5} samples gathered from 14 diverse sites across 4 different continents and utilizing 5 different endpoints. Indeed, sampling from so many locations worldwide is quite exceptional, and the workload is substantial. However, there are deficiencies in novelty and the rationality of experimental design. Overall, the manuscript does not seem to meet the publishing standards of Nature Communications. The lower quality of the manuscript has also impacted my reading experience, with the occurrence of frequent low-level errors.

Specific Comments

- 1) We note that the authors have actually evaluated the cytotoxicity of water-soluble PM extracts. Considering the highly complex composition of PM, there are insoluble components in addition to water-soluble extracts. Therefore, how can water-soluble PM extracts substitute for the overall toxicity of complete PM? This to some extent limits the accuracy of the toxicity results assessment.
- 2) The author alternately uses "PM" and "PM_{2.5}" when describing particulate matter. Please use "PM_{2.5}" consistently throughout the entire text.
- 3) The author employs oxidative potential, cell death, and ROS to assess PM cytotoxicity. However, oxidative potential and ROS should have a certain correlation, for instance, hydroxyl radical potential is also a type of ROS. What criteria did the author use to distinguish between these indicators?
- 4) The author indicates in this work that the relationship between PM_{2.5} and health effects is not solely driven by PM mass, and the role of chemical composition, which drives its intrinsic toxicity, cannot be ignored. This conclusion appears too simplistic, lacking depth, and lacks novelty. Many studies have already demonstrated that the chemical composition of PM_{2.5} does indeed influence its health effects.

Reviewer #2 (Remarks to the Author):

Some comments and questions as follows :

Overall, the time-dependent effect for each indicator has not been considered thoroughly. For example, in the Cellular ROS and cytotoxicity measurements, why the exposure time choose 24 hour. Do you have considered the difference of Cellular ROS and cytotoxicity in other exposure time? Also in the Acellular OP Measurements, the time-dependent effect also need to be considered. Multipoints slope method may be better to the endpoint method.

For the cellular ROS measurement, after incubation 24 h the cells may have dead, did the ROS can be detect in dead cells? How do the dead cells affect the cellular ROS?

In this study, a lung epithelial cell line, A549 have been chosen, how about other cell lines? In the epidemiology study, Particles have been proved to be greatest damage to cardiovascular diseases, have you considered cardiovascular cells?

In this paper, you only test toxicity of the water-soluble extracts, how about the water-insoluble parts? Do you consider the PM extracts extracted from other solvents?

Why not choose inflammatory factors as the indicators?

The discussion between toxic indicators and PM chemical components need be strengthened.

Reviewer #3 (Remarks to the Author):

Salana et al. present comprehensive aerosol mass, toxicity, and chemical composition measurements performed on a large set of aerosol filter samples collected over a broad geographical range. This is an important and valuable dataset because it allows for direct comparison between the samples collected in different locations, which is often not possible due to

the non-standardized nature of the aerosol toxicity measurements performed by different research groups. This comparison allows the authors to reach the conclusion that the relationships between their measured health endpoints and PM mass is different in different locations. This is an important observation that has several interesting implications as discussed by the authors. I have a few specific comments that I believe require revision and/or clarification.

One important methodological detail that is currently missing is a description of how the PM_{2.5} mass concentrations were measured. Since part of the analysis hinges on the variability in PM_{2.5} mass concentrations relative to those in the measured health endpoints it would also be important to report some uncertainty estimates for the different measurements.

The focus of the paper is on the relationship between the 5 aerosol toxicity metrics and PM_{2.5} mass. In my opinion this is a good choice since a clear message emerges from the paper. Nevertheless, many experts will also be interested in the relationships between the 5 toxicity metrics themselves, since the development of such metrics is an active area of research. Perhaps the authors could add a heatmap or scatterplot matrix to the supplementary information to show these relationships, while noting that a detailed discussion of all the differences would be beyond the scope of the paper.

L46: Typo? 'a dearth', rather than simply 'dearth'.

L84: To perform a study with such broad geographical coverage it is necessary to use offline filter samples. However, this brings the limitation that the toxicity measurements will not capture short-lived ROS and OP-relevant species. I think it is worth briefly discussing this limitation, especially since the samples had variable storage times.

L131: Could these differences be partly due to the use of CoV as a metric, given this partially depends on sample size, which is more than an order of magnitude higher for the Midwest US than India? The authors might consider ways of demonstrating that sample size is not a critical factor when making comparisons across the different regions.

L138: One other potential issue with the use of CoV as a statistical metric is inflated values due to division by means that approach zero. I am wondering if this could be an issue for some of the GSHv CoV values, given the GSHv measurements sometimes approach zero, and even more so for the element CoV values shown in Fig. S7, since I assume some of the elements were present in very low concentrations. The authors could look at additional metrics of variability to support their observation that the elemental concentrations varied more widely than the corresponding total mass concentrations.

L151: For completeness it should be mentioned that there are chemical species that were not resolved by the measurements that could be influencing the observed variability in the measured health endpoints (e.g. different organic aerosol fractions within the total WSOC).

L215: The non-linearity is mainly driven by the 1 set of India samples with generally lower intrinsic toxicity. To some extent this is an unavoidable limitation of the sample size (385 samples is already a lot relative to other studies). Nevertheless I think it argues against the usefulness of purely data-driven non-linear fits. For example, do the authors consider the cubic fits or CDv and OHv against PM_{2.5} to be physically reasonable? What are the risks of extrapolating these relationships beyond their fitted ranges? I wonder if a better approach would be to assume a specific non-linear functional form (e.g. a log function) that has some basis in the literature and/or theory, and then to compare that against a standard linear fit. Are the authors aware of any non-linear functional forms that have been proposed in previous studies that could perform this role?

Related to the above point, a clear point that emerges from Fig. 4 is the need for more measurements at medium and high PM concentrations to better constrain the non-proportional relationships between these metrics and PM_{2.5} mass. Perhaps the authors could discuss this point as a goal for future studies.

Figure 4: The smallest data points (lowest intrinsic values) are difficult to see. The size of these

data points could be increased for better visualization, perhaps while reducing the number of intrinsic categories (e.g. from >6 to ~ 3 or 4) so that the largest marker size is not too large.

Response Document

We thank all three reviewers for their feedback and review of our manuscript and for providing us the opportunity to address their comments. We have responded to all the comments individually below and revised the manuscript as per the reviewers' suggestions. All the responses are highlighted in red font and all the changes in the manuscript are italicized and highlighted in red font. Please note that the line and page numbers we provided in the point-by-point response refer to the annotated copy of the revised manuscript that tracks deletions and additions unless stated otherwise.

Reviewer #1 (Remarks to the Author):

In this manuscript, the authors assess the connection between particulate matter and oxidative potential by employing 385 PM_{2.5} samples gathered from 14 diverse sites across 4 different continents and utilizing 5 different endpoints. Indeed, sampling from so many locations worldwide is quite exceptional, and the workload is substantial. However, there are deficiencies in novelty and the rationality of experimental design. Overall, the manuscript does not seem to meet the publishing standards of Nature Communications. The lower quality of the manuscript has also impacted my reading experience, with the occurrence of frequent low-level errors.

We thank the reviewer for their appreciation of the sampling and workload undertaken in this study. We apologize for the low-level errors. The manuscript has been substantially revised and proof-read to remove such errors. We hope the revised manuscript will provide a much improved reading experience to the reviewer.

Specific Comments

- 1) **We note that the authors have actually evaluated the cytotoxicity of water-soluble PM extracts. Considering the highly complex composition of PM, there are insoluble components in addition to water-soluble extracts. Therefore, how can water-soluble PM extracts substitute for the overall toxicity of complete PM? This to some extent limits the accuracy of the toxicity results assessment.**

Response:

We agree with the reviewer that the composition of PM_{2.5} is highly complex containing both water-soluble and insoluble components. As mentioned in our manuscript, we have measured only water-soluble oxidative potential (OP) and cytotoxicity. This decision was driven by several unavoidable factors. Devising suitable extraction protocols for offline filters that could replicate the fate of inhaled particles in the most physiologically relevant manner is a topic of ongoing research. However, currently there is no universally accepted protocol for total PM_{2.5} extraction for OP and toxicity measurements. A method for measuring total OP was proposed by Gao et al¹, in which the entire filter was submerged in the reaction vial so that all the particles in the filter could participate in the oxidative stress reaction. Although this method is suitable for acellular OP measurements, it cannot be applied to cellular OP and cytotoxicity measurements as filter fibers

could themselves be toxic to the cells². Thus, application of this method will complicate the computation of PM_{2.5}'s actual contribution to cytotoxicity.

So far, two extraction procedures for extracting the PM_{2.5} from filters have been widely used: water extraction and organic solvent extraction. Preparation of water extracts is rather straightforward. However, there are several issues associated with the extraction of PM_{2.5} using organic solvents. First, there are several organic solvents [e.g., methanol, dichloromethane (DCM), hexane, acetone, and acetonitrile, etc.], typically used for extracting water-insoluble PM_{2.5} components²⁻⁴. These solvents differ widely in their polarities, resulting in the extraction of different PM_{2.5} components² leading to different OP responses. For example, it has been shown that the solubility of brown carbon (BrC) in methanol could be significantly different compared to its solubility in DCM and this difference could vary depending on the source of BrC⁵. Second, any organic solvent used to extract the PM_{2.5} needs to be evaporated (e.g., using N₂ gas or heating in a rotary evaporator)^{2,6,7}, which could also result in the loss of labile PM_{2.5} components⁸, thus reducing the advantages purported by these solvent extraction procedures. Third, the influence of these solvents on modifying cellular responses has not yet been adequately quantified. For example, it has been reported in a previous study that interactions between DCM and the filter material could substantially amplify the cytotoxicity and thus misrepresent the actual toxicity of PM_{2.5} components². Finally, the physiological relevance of using such organic solvents for toxicity studies has not yet been established. It is possible that the solubility of water-insoluble compounds in lung fluids could vastly differ from their solubility in organic solvents. Thus, by using such organic solvents we might be overestimating the contribution of water-insoluble species to the overall toxicity of PM_{2.5}.

We fully agree with the reviewer that water-soluble extracts do not represent the overall toxicity of PM_{2.5}. However, considering all these issues associated with the extraction of water-insoluble PM_{2.5} components and the measurement of their cellular toxicity effects, we decided to choose water-soluble extracts, so that the comparison between cellular and acellular OP or cytotoxicity is reasonable. Moreover, water-soluble extraction has been employed in innumerable OP-related studies in the past⁹⁻¹⁴ and the OP measured on such extracts has even been shown to be associated with several diseases such as asthma/wheeze and congestive heart failure¹⁵ and ischemic heart disease¹⁶.

Considering the reviewer's concern, we have added a few sentences to address this limitation in the "Limitations and implications" subsection of the revised annotated manuscript (Page 15, Lines 299 to 307). We have reproduced the discussion here for the convenience of the reviewer:

"Second, our study focused only on water-soluble extracts of PM_{2.5}. Water-insoluble species of PM_{2.5} have also been shown to contribute to PM_{2.5} toxicity^{2,7,17}, and thus water-soluble fraction of the PM_{2.5} used in our study can be considered as the lower limit, as it does not fully substitute for the overall toxicity of PM_{2.5}. Note, our choice of water-soluble fraction was driven by the lack of a standardized protocol to measure total OP or cytotoxicity of PM_{2.5} that is equally applicable to both acellular and cellular assays. Although, several solvents (e.g., methanol, dichloromethane, hexane, acetone, and acetonitrile) have been suggested to extract the water-insoluble fraction of

the PM_{2.5}²⁻⁴, the adequacy of these solvents to retain PM_{2.5} chemical composition, which is physiologically relevant for the cellular exposure has not been tested.”

- 2) **The author alternately uses “PM” and “PM_{2.5}” when describing particulate matter. Please use “PM_{2.5}” consistently throughout the entire text.**

Response:

We apologize for the inconsistency. We have now used “PM_{2.5}” throughout the text.

- 3) **The author employs oxidative potential, cell death, and ROS to assess PM cytotoxicity. However, oxidative potential and ROS should have a certain correlation, for instance, hydroxyl radical potential is also a type of ROS. What criteria did the author use to distinguish between these indicators?**

Response:

In our study we have used acellular OP measurements to quantify the oxidative potential of PM_{2.5} in cell-free conditions and cellular ROS measurements to quantify the oxidative potential of PM_{2.5} in A549 cells. For acellular OP measurements, we used three different assays: DTT depletion, GSH depletion and OH• generation, and for cellular ROS, we used the DCFH-DA assay. All these assays represent different aspects of the oxidative stress, as explained below.

Dithiothreitol or DTT depletion is primarily used as a proxy for the ability of PM_{2.5} to deplete the cellular reductant NADPH (nicotinamide adenine dinucleotide phosphate)^{18,19} which plays a key role in the energy production and metabolism of the cells by reducing oxygen to the superoxide anion²⁰. Glutathione or GSH is the most abundant antioxidant present in the cells, which plays an important role in protecting the cell against oxidative stress and performs other key cellular activities such as redox signaling and cell proliferation²¹. Measuring GSH depletion provides an estimate of the antioxidant depletion ability of PM_{2.5}. OH• is a highly reactive oxygen species with a very short half-life (10⁻⁹s)²² that could damage critical cellular organelles (e.g., cytoplasmic membrane) and molecules (e.g., DNA)²³. Measuring the ability of PM_{2.5} to generate OH• is thus essential to assess the capability of the inhaled PM_{2.5} to cause damage to the cytoplasmic membrane and DNA of the cells in respiratory tract.

The DCFH-DA probe used in this study to measure cellular ROS response in A549 cells is a widely used non-specific ROS probe, which reacts with a broad range of oxygen species (e.g., H₂O₂, OH•, ROO•, [•]O₂⁻, etc.), providing a general assessment of the overall redox state of the cells rather than a quantitative estimate of the specific ROS²³. We agree with the reviewer that hydroxyl radical is a type of ROS, but it is only a subset of the overall ROS induced by the PM_{2.5}. Therefore, it is not reasonable to expect an explicit correlation between hydroxyl radical and total ROS because the proportion of different ROS generated by PM_{2.5} could vary widely among different PM_{2.5} samples. Similarly, GSH and DTT depletion measurements provide antioxidant depletion ability of PM_{2.5} which may or may not be correlated with ROS response of the cells as shown by several previous studies²⁴⁻²⁶. Thus, the different OP endpoints used in this study refer to different aspects of

oxidative stress, which is difficult to be captured by a single acellular or cellular assay, necessitating the use of multiple endpoints to comprehensively assess PM_{2.5} toxicity.

Based on the reviewer’s suggestion, we have attempted a correlation analysis among all OP and cytotoxicity endpoints. The heatmap with the correlation coefficients (Pearson’s r) for different regions is shown in the Supplementary Fig. 9 (reproduced below for the convenience of the reviewer). The figure shows that there was no appreciable correlation among most of the intrinsic endpoints, except for the correlation between OP^Cm and CDm ($r > 0.5$). The correlations among extrinsic endpoints were better, although no consistent pattern was observed across different regions. For example, CDv showed a strong correlation with OP^Cv ($r > 0.6$) and a moderate correlation with almost all acellular OP ($0.4 < r < 0.6$) in most of the regions, except Midwest US. On the other hand, OP^{GSH}v and OP^{OH}v showed a weak correlation ($r < 0.4$) with other endpoints at most of the sites except West Midlands ($r > 0.4$). This inconsistent pattern of correlation could also be due to the fact that various cellular and acellular OP endpoints are shown to be driven by different chemical species. For example, OP^{DTT} has been shown to be mostly associated with WSOC and metals such as Cu, Mn, Zn and Fe in previous studies^{7,27}. OP^{GSH} has been shown to be associated with Fe, Cu, Pb and Al⁶, while OP^{OH} seems to be driven mostly by Fe and Cu²⁸. Cellular OP has also been associated with a number of different species such as Cu, Mn, Ni, Cr, As, Pb, and WSOC^{29,30}. Such a diversity in the relationship between OP and chemical components underlines the complex toxic mechanisms triggered by PM_{2.5}, thus necessitating the use of different assays. In fact, the use of multiple endpoints (both cellular and acellular) to evaluate oxidative stress using such a wide array of samples is one of the strengths of our work as only a handful of studies have evaluated PM_{2.5} OP in such a comprehensive manner.

Supplementary Fig. 9: Heatmaps showing correlations (Pearson’s r) among different OP and cytotoxicity endpoints for different regions. Panel a) shows correlations among intrinsic endpoints and b) shows correlations among extrinsic endpoints.

We apologize for the confusion that might have arisen due to the nomenclature used in the manuscript. To clarify the difference between cell-based and cell-free OP measurements, we have replaced “cellular ROS” with “cellular OP” in the revised manuscript. Therefore, we now use the term “acellular OP (OP^{DTT}, OP^{GSH} and OP^{OH})” to refer to all OP measurements conducted in cell-free conditions and “cellular OP (OP^C)” to refer to OP measurements (through DCFH-DA) conducted in A549 cells.

- 4) **The author indicates in this work that the relationship between PM_{2.5} and health effects is not solely driven by PM mass, and the role of chemical composition, which drives its intrinsic toxicity, cannot be ignored. This conclusion appears too simplistic, lacking depth, and lacks novelty. Many studies have already demonstrated that the chemical composition of PM_{2.5} does indeed influence its health effects.**

Response:

We disagree with the reviewer’s comment on the lack of novelty in our manuscript. We agree that several studies, some of which have already been cited in our manuscript, have demonstrated that chemical composition of PM_{2.5} influences its health effects. However, what is lacking in those previous studies is the mechanism on how this variation in the chemical composition results in variable health effects. Oxidative stress or OP is the property of PM_{2.5}, which has been hypothesized as the bridge between chemical composition and health effects, with evidences coming from both laboratory studies showing the linkages of chemical composition of PM_{2.5} with OP^{13,14,27,28,31–33}, and clinical/epidemiological studies to link OP with the health effects^{15,16,34–36}. For the first time, we showed a similar non-linear relationship between OP (or cytotoxicity) and PM_{2.5} mass concentrations, as observed between the health effects vs. PM_{2.5} in recent epidemiological studies. This would have been impossible without the extensive measurements we have made in different geographical regions with different emission settings and meteorology. Thus, as stated on line 216 of the original manuscript, the novelty of our manuscript is to provide an important mechanistic basis for the observed non-linearity in the relationship between PM_{2.5} mass concentrations and health effects.

Line 216 of the original manuscript:

“These results, which are strikingly consistent with newer epidemiological studies demonstrating a supralinear relationship between PM_{2.5} mass and mortality observed at lower PM_{2.5} concentrations⁵⁰ and the flattening of the CR curve at higher PM_{2.5} concentrations⁵¹, provide an important mechanistic basis for the nonlinear relationship between PM_{2.5} mass and health effects.”

Interestingly, despite numerous studies showing a better relationship of health effects with OP than PM_{2.5} mass, the studies measuring OP of PM_{2.5} on large spatial extents are very limited. There are few studies which have demonstrated that the variation in PM_{2.5} mass concentrations does not correspond to the variation in OP^{17,37}. However, these studies have been limited to specific geographical regions such as Midwest US¹⁷ and Europe³⁷. Therefore, it is not yet clear if such trends are specific to a given region due to biases from specific chemical composition of PM_{2.5} in that region. For example, if the main drivers (i.e., PM_{2.5} chemical constituents) of OP correlate

well with PM_{2.5} mass in a region, mass could seem to capture the PM_{2.5} toxicity or OP well, but this relationship will change when OP drivers no longer correlate with mass.

In our current study, we have demonstrated that such trends (i.e., disparity of PM_{2.5} OP or cytotoxicity vs. mass) are not specific to certain regions or a specific toxicity assay. Rather, the variations in PM_{2.5} mass in general are almost always lower than the variations in its OP. This finding is also significantly important because it shows the inability of PM_{2.5} mass to adequately capture the variations in toxicity of PM_{2.5} and thus justifies the importance of intrinsic toxicity measurements in understanding the health effects of PM_{2.5}. We are not aware of any other study which has demonstrated the importance of developing regional concentration-response (CR) curves through intrinsic toxicity and/or OP measurements, as made possible through our extensive measurements at 14 sites across the world.

To summarize, our conclusions go beyond simply stating that the health effects of PM_{2.5} are dependent on its chemical composition. Rather, we provide the mechanistic basis for this statement, underlining the importance of including toxicity measurements in epidemiological studies.

Reviewer #2 (Remarks to the Author):

Some comments and questions as follows:

- 1) Overall, the time-dependent effect for each indicator has not been considered thoroughly. For example, in the Cellular ROS and cytotoxicity measurements, why the exposure time choose 24 hour. Do you have considered the difference of Cellular ROS and cytotoxicity in other exposure time? Also in the Acellular OP Measurements, the time-dependent effect also need to be considered. Multipoints slope method may be better to the endpoint method.**

Response:

We apologize if there is any confusion, but we have considered time-dependent effects, using multipoint slope method as suggested by the reviewer for all acellular OP measurements. We have used the instrument SAMERA to measure acellular OP. This instrument indeed uses a multipoint slope method in which the concentrations of DTT, GSH, and OH• are measured at designated time intervals, and then uses these measurements to calculate their depletion or generation rates over the entire time period. This procedure is described in detail in a previous publication (Yu et al. 2020) from our research group³⁸.

Briefly, the designated time intervals for various acellular OP assays are as follows:

1. For OP^{DTT}, the absorbance of 2-nitro-5-thiobenzoic acid (TNB, a yellow-colored complex formed as a result of the reaction of DTNB with residual DTT in the measurement vial) was measured at 5, 17, 29, 41, and 53 min.
2. For OP^{GSH}, fluorescence of GS-OPA was measured at 5, 24, 43, 62, and 81 min.
3. For OP^{OH}, fluorescence of 2-OHTA was measured at 10, 29, 48, 67, and 86 min.

The calibration curves to quantify GSH, OH•, and DTT are prepared by measuring initial fluorescence (for GSH, OH•) and absorbance (for DTT) intensity of different known concentrations of GSH, OH•, and DTT, following the same protocol as described in the manuscript. The consumption rate [$\mu\text{M}/\text{min}$ (in case of GSH and DTT) and generation rate (in case of OH•)] are then derived from these calibration curves.

Considering the reviewer's concern that it might not have been clear from the original manuscript, we have now modified the description of acellular assays in the "Methods" section of the revised annotated manuscript (Page 24, Line 506 to Page 25, Line 532) to briefly describe the multipoint slope method used in this study for acellular OP measurements. The modified section is reproduced here for the convenience of the reviewer:

"DTT consumption rate was measured using the 5,5'-dithiobis- (2-nitrobenzoic acid) (DTNB) method as described in our previous publication³⁹. Briefly, DTNB and a small aliquot from the RV containing a mixture of PM_{2.5} extract and DTT were added to a measurement vial (MV). The DTNB reacts with residual DTT to form a yellow-colored compound called 2-nitro-5-thiobenzoic acid (TNB). TNB was then diluted using DI and passed through a liquid waveguide capillary cell (LWCC-410 3100; World Precision Instruments, Inc., Sarasota, FL, USA), where the absorbance at 412 nm and 600 nm (background) was measured by the spectrophotometer (Ocean Optics;

Dunedin, FL, US). *This process was repeated at time intervals of 5, 17, 29, 41, and 53 min to obtain the PM_{2.5}-catalyzed DTT decay rate. GSH depletion rate was measured using the o-phthaldialdehyde (OPA) method³⁸. In this method, a small aliquot from the RV containing PM_{2.5} extract and SLF was withdrawn at time intervals of 5, 24, 43, 62, and 81 min, and transferred to a MV along with OPA. GSH reacts with OPA to form a fluorescent product called GS-OPA, and the fluorescence was measured at an emission wavelength of 427 nm (excitation wavelength = 310 nm) to estimate the residual GSH concentration. Finally, OH• generation rate was measured using the 2-OHTA method³⁸. In this method, disodium terephthalate (TPT) is added to the RV containing SLF and PM_{2.5} extract to capture the OH• generated during the reaction of PM_{2.5} with the antioxidants contained in SLF. The reaction between TPT and OH• produces a fluorescent product: 2-OHTA, which was withdrawn from the RV at time intervals of 10, 29, 48, 67, and 86 min and diluted with DI in a MV. The diluted 2-OHTA was then passed through the flow cell of the spectrofluorometer (Fluoromax-4, Horiba Scientific, Edison, NJ, USA) to measure its fluorescence (excitation: 310 nm; emission: 427 nm). The instrument was calibrated with known standards of 2-OHTA (0-200 nM) and a yield factor of 0.35 (formation of 2-OHTA from OH•) was applied to determine the concentration of OH•. The slopes of the DTT, GSH, and OH• concentration vs. time curves were then used to determine the consumption (in case of DTT and GSH) and generation (in case of OH•) rates (μM/min) in various OP assays.”*

For cellular ROS and cytotoxicity measurements, our choice of 24 h was driven by two factors: 1. the stability of DCFH-DA probe used in cellular OP assay; and 2. the consistency between the duration for cellular OP and cytotoxicity experiments so that their results are comparable. For the 1st factor, we conducted some initial experiments to assess the stability of DCFH-DA working solution at different time points and chose the most optimal exposure duration for our experiments. A working solution of DCFH-DA (450 μM) was prepared, and its absolute fluorescence was measured at various time points: 0, 1, 3, 4, 5, 6, 8, 12, 25, 48 and 72 h. Concurrently, DCFH was prepared by deacetylating DCFH-DA (450 μM) using the method described in Reiniers et al.⁴⁰ and mixed with 100 μM H₂O₂ (working as a surrogate for the ROS generated by the cells). The fluorescence of this mixture was also measured at the same time points (i.e., 0, 1, 3, 4, 5, 6, 8, 12, 25, 48, and 72 h). It was observed that the fluorescence of DCFH-DA was relatively negligible until 25 h and started to gradually increase thereafter. However, the absolute fluorescence of DCFH and H₂O₂ mixture attained its maximum value within 8 h and stayed constant thereafter, indicating that the entire DCFH had reacted with H₂O₂ within 8 h to form DCF. Therefore, it can be concluded that the fluorescence of DCF formed due to the reaction between DCFH and ROS would remain stable till 24 h or more. However, the working solution of DCFH-DA could become unstable and undergo autoxidation for an incubation period of more than 24 h. Thus, any measurement of ROS after 24 h using this method may not be reliable. Therefore, the maximum duration for cellular OP experiment was limited to 24 h in our study.

Although, we could have measured cellular OP and cytotoxicity for shorter exposure durations, most cytotoxicity experiments involving PM_{2.5} use a minimum exposure duration of 24 h. We wanted to keep the same duration for cellular OP and cytotoxicity experiments, so that their results are comparable (2nd factor). Therefore, 24 h exposure was chosen to keep a consistent exposure duration for both assays. However, we acknowledge that both cellular ROS and cytotoxicity could

show time dependent response and this response could vary with the chemical composition of PM_{2.5}. To address the reviewer’s concern, we have added this limitation to the “Limitations and implications” subsection of the revised annotated manuscript (Page 15, Lines 316 to 319):

“We also acknowledge that although we measured the time-dependent responses of acellular assays, given the laborious protocols of cellular assays, we followed the conventional method based on measuring cellular responses only at 24 h, which could underestimate some of these endpoints.”

We have added the following lines (Page 20, Line 429 to Page 21, Line 436) to the revised annotated manuscript to highlight the reason for choosing 24 h as the exposure duration:

“In the second step, PM_{2.5} water extract (78 μL), working solution of DCFH-DA (22 μL; DCFH-DA preparation details are given in the Supplementary Method 1 in SI) and culture medium (100 μL), were added and the cells were incubated in the dark for 24 h. The exposure duration of 24 h was chosen because the working solution of DCFH-DA was found to be stable till at least 25 h in our initial experiments. Details of the experiment showing the time-dependent absolute fluorescence of DCFH-DA are provided in Supplementary Method 2 and Supplementary Fig. 1 in SI.”

We have also added the following details about the experiments conducted on the stability of DCFH-DA in the SI of the revised manuscript:

Supplementary Fig. 1: Variation in the absolute fluorescence of DCFH-DA over time. The blue line represents the fluorescence of DCFH treated with H₂O₂ whereas the red line represents the fluorescence of the working solution of DCFH-DA without any treatment.

“Supplementary Method 2: Stability of the working solution of DCFH-DA

To determine the stability of DCFH-DA for cellular OP measurements, the working solution of DCFH-DA (450 μM) was prepared, and its absolute fluorescence was measured at various time points: 0, 1, 3, 4, 5, 6, 8, 12, 25, 48 and 72 h. Concurrently, DCFH was prepared by deacetylating

DCFH-DA (450 μ M) using the method described in Reiniers et al.⁴⁰ and mixed with 100 μ M H_2O_2 (working as a surrogate for the ROS generated by the cells). The fluorescence of this mixture was also measured at the same time points (i.e., 0, 1, 3, 4, 5, 6, 8, 12, 25, 48, and 72 h). As shown in Supplementary Fig. 1, the absolute fluorescence of DCFH-DA was relatively negligible until 25 h and started to gradually increase thereafter. However, the fluorescence of the DCFH and H_2O_2 mixture attained its maximum fluorescence within 8 h and stayed constant thereafter, indicating that the entire DCFH had reacted with H_2O_2 to form DCF. Therefore, it can be concluded that the fluorescence of DCF formed due to the reaction between DCFH and ROS remains stable till at least 24 h. However, the working solution of DCFH-DA could become unstable and undergo autoxidation for an incubation period of more than 24 h.”

- 2) For the cellular ROS measurement, after incubation 24 h the cells may have dead, did the ROS can be detect in dead cells? How do the dead cells affect the cellular ROS?**

Response:

Yes, in our modified DCFH-DA method, the ROS generated in the dead cells are also detected. We agree with the reviewer that the traditional cellular OP protocol, which first involves incubating the cells with the toxicant for 24 h (in absence of DCFH-DA), followed by incubating them with DCFH-DA for 30-60 min^{41,42}, fails to capture the ROS response of dead cells during first incubation period. This is because DCFH-DA needs esterase enzymes to form DCFH and these esterase enzymes are present only in viable cells⁴³, therefore they do not capture any ROS that may have been generated in the dead cells. However, we have modified the DCFH-DA method based on a few recent studies^{44,45} to address this issue. Our method ensures that the cells are preloaded with DCFH-DA to capture the total ROS generated during the entire 24 h exposure period. In our protocol, by exposing the cells to a mixture of DCFH-DA, culture medium and $PM_{2.5}$ extract, we can capture both intracellular and extracellular ROS causing cytotoxicity throughout the exposure duration. Therefore, even though the cells are dead, most of the ROS in the cells would have been captured by DCFH-DA.

- 3) In this study, a lung epithelial cell line, A549 have been chosen, how about other cell lines? In the epidemiology study, Particles have been proved to be greatest damage to cardiovascular diseases, have you considered cardiovascular cells?**

Response:

We agree with the reviewer that $PM_{2.5}$ has been associated with cardiovascular diseases in epidemiological studies and evaluating the effect of $PM_{2.5}$ toxicity on cardiovascular cells may help establish the mechanistic basis of such studies. However, our choice of a lung cell line was driven by the fact that inhaled ambient particles are first intercepted by cells in the respiratory tract. The toxicity caused in these cells and the associated damage to lungs can also translate to the other parts of the body such as the cardiovascular region. Therefore, to understand the genesis of cardiovascular diseases, it is important that we have a firm understanding of the effect of $PM_{2.5}$ on the pulmonary region first. We chose A549 as it is a human alveolar basal epithelial cell line which is representative of the cells responsible for the diffusion of substances, such as water and electrolytes, across the alveoli of lungs⁴⁶. The cells in this region play a crucial role in preventing

inflammation⁴⁷ and maintaining the normal lung architecture by renewing other types of alveolar cells⁴⁸. It is possible that the inhaled particles can translocate to the cardiovascular region^{49,50}, but it is the alveolar region, which facilitates the entrainment of particles and their constituents into other regions of our body by crossing the blood-air barrier⁵¹. Therefore, to understand the effect of PM_{2.5} on any other organs of our body, it is crucial that we understand its effect on the alveoli first. A549 is a well-established and one of the most widely used cell lines in PM_{2.5} studies⁵²⁻⁵⁵.

We have added the following sentences in the “Methods” section of the revised annotated manuscript (Page 20, Lines 413 to 419) to justify the choice of A549 cells in our study:

“A549 is one of the most widely used cell lines in PM_{2.5} toxicological studies⁵²⁻⁵⁴ and is representative of the cells responsible for the diffusion of substances, such as water and electrolytes, across alveoli of the lungs. The cells in this region play a crucial role in preventing inflammation⁴⁷, and maintaining the normal lung architecture by renewing other types of alveolar cells⁴⁸. Moreover, the alveolar region facilitates the entrainment of particles and their constituents into other regions of our body by crossing the blood-air barrier⁵¹, making A549 a suitable choice for our study.”

We also acknowledge that there are cell lines which are representative of other parts of the pulmonary and cardiovascular region that could be used as a model for testing PM_{2.5} toxicity. We have added the following sentences in the “Limitations and implications” subsection (Page 15, Lines 307 to 313) of the revised annotated manuscript:

“The choice of our cell line and the endpoints could also limit implication of cellular toxicity results. Although, A549 is a widely used alveolar epithelial cell line relevant to alveolar exposure to PM_{2.5}, its responses cannot be equated to other cell lines such as BEAS-2B, 16-HBE14o, Calu-3 (relevant to broncho tracheal region exposure to PM_{2.5}), and pulmonary and cardiovascular cell lines [e.g., THP-1 (macrophages), HEK-293, HMVEC-L and HULEC-5a (human lung microvascular endothelial cells), and H9C2].”

4) In this paper, you only test toxicity of the water-soluble extracts, how about the water-insoluble parts? Do you consider the PM extracts extracted from other solvents?

Response:

We thank the reviewer for bringing up the point about other solvents. This comment is similar to the 1st comment of Reviewer #1. We have reproduced our response to the comment here for the convenience of the reviewer:

“We agree with the reviewer that the composition of PM_{2.5} is highly complex containing both water-soluble and insoluble components. As mentioned in our manuscript, we have measured only water-soluble oxidative potential (OP) and cytotoxicity. This decision was driven by several unavoidable factors. Devising suitable extraction protocols for offline filters that could replicate the fate of inhaled particles in the most physiologically relevant manner is a topic of ongoing research. However, currently there is no universally accepted protocol for total PM_{2.5} extraction for OP and toxicity measurements. A method for measuring total OP was proposed by Gao et al¹., in which the entire filter was submerged in the reaction vial so that all the particles in the filter

could participate in the oxidative stress reaction. Although this method is suitable for acellular OP measurements, it cannot be applied to cellular OP and cytotoxicity measurements as filter fibers could themselves be toxic to the cells². Thus, application of this method will complicate the computation of PM_{2.5}'s actual contribution to cytotoxicity.

So far, two extraction procedures for extracting the PM_{2.5} from filters have been widely used: water extraction and organic solvent extraction. Preparation of water extracts is rather straightforward. However, there are several issues associated with the extraction of PM_{2.5} using organic solvents. First, there are several organic solvents [e.g., methanol, dichloromethane (DCM), hexane, acetone, and acetonitrile, etc.], typically used for extracting water-insoluble PM_{2.5} components²⁻⁴. These solvents differ widely in their polarities, resulting in the extraction of different PM_{2.5} components² leading to different OP responses. For example, it has been shown that the solubility of brown carbon (BrC) in methanol could be significantly different compared to its solubility in DCM and this difference could vary depending on the source of BrC⁵. Second, any organic solvent used to extract the PM_{2.5} needs to be evaporated (e.g., using N₂ gas or heating in a rotary evaporator)^{2,6,7}, which could also result in the loss of labile PM_{2.5} components⁸, thus reducing the advantages purported by these solvent extraction procedures. Third, the influence of these solvents on modifying cellular responses has not yet been adequately quantified. For example, it has been reported in a previous study that interactions between DCM and the filter material could substantially amplify the cytotoxicity and thus misrepresent the actual toxicity of PM_{2.5} components². Finally, the physiological relevance of using such organic solvents for toxicity studies has not yet been established. It is possible that the solubility of water-insoluble compounds in lung fluids could vastly differ from their solubility in organic solvents. Thus, by using such organic solvents we might be overestimating the contribution of water-insoluble species to the overall toxicity of PM_{2.5}.

We fully agree with the reviewer that water-soluble extracts do not represent the overall toxicity of PM_{2.5}. However, considering all these issues associated with the extraction of water-insoluble PM_{2.5} components and the measurement of their cellular toxicity effects, we decided to choose water-soluble extracts, so that the comparison between cellular and acellular OP or cytotoxicity is reasonable. Moreover, water-soluble extraction has been employed in innumerable OP-related studies in the past⁹⁻¹⁴ and the OP measured on such extracts has even been shown to be associated with several diseases such as asthma/wheeze and congestive heart failure¹⁵ and ischemic heart disease¹⁶.

Considering the reviewer's concern, we have added a few sentences to address this limitation in the "Limitations and implications" subsection of the revised annotated manuscript (Page 15, Lines 299 to 307). We have reproduced the discussion here for the convenience of the reviewer:

"Second, our study focused only on water-soluble extracts of PM_{2.5}. Water-insoluble species of PM_{2.5} have also been shown to contribute to PM_{2.5} toxicity^{2,7,17}, and thus water-soluble fraction of the PM_{2.5} used in our study can be considered as the lower limit, as it does not fully substitute for the overall toxicity of PM_{2.5}. Note, our choice of water-soluble fraction was driven by the lack of a standardized protocol to measure total OP or cytotoxicity of PM_{2.5} that is equally applicable to both acellular and cellular assays. Although, several solvents (e.g., methanol, dichloromethane,

hexane, acetone, and acetonitrile) have been suggested to extract the water-insoluble fraction of the PM_{2.5}²⁻⁴, the adequacy of these solvents to retain PM_{2.5} chemical composition, which is physiologically relevant for the cellular exposure has not been tested.”

5) Why not choose inflammatory factors as the indicators?

Response:

The reviewer raises a valid point about the importance of inflammatory factors as indicators of PM_{2.5} toxicity. However, there are several inflammatory factors such as CXCL8, IL-1 α , IL-1 β , COX2, and TNF- α which are triggered by PM_{2.5}⁵⁶, and it is not yet established which one of these is most relevant to PM_{2.5}-induced toxicity. Moreover, given the constraint we had related to the PM_{2.5} mass loadings on these filters (minimum mass = 1200 μ g), it was not possible for us to evaluate all these inflammatory factors for all of the samples along with the acellular OP measurements. Even the conducted analyses (i.e., acellular and cellular OP, and cytotoxicity) consumed some of the filters completely. Therefore, the choice of cytotoxicity and ROS response was driven by the need for bulk parameters, which could better represent the PM_{2.5}-induced toxicity. For example, the cytotoxicity endpoint (i.e., cell death) measured in our study probably incorporates some of these intermediate inflammatory responses and is a more wholesome metric than the inflammatory markers.

Considering the reviewer’s concern, we have added the following additional sentences (Page 15, Lines 313 to 316) in the “Limitations and implications” subsection of the revised annotated manuscript acknowledging that there might be other indicators of PM_{2.5} toxicity that were not measured in our study:

“Measurement of cellular responses other than cellular OP and cytotoxicity, such as inflammatory cytokines, gene expressions and specific type of cell death (e.g., necrosis, apoptosis and autophagy) could provide more valuable insights into the toxicity mechanisms triggered by PM_{2.5}.”

6) The discussion between toxic indicators and PM chemical components need be strengthened.

Response:

We thank the reviewer for this suggestion. However, we intentionally avoided the discussion on the relationship between different toxic indicators and PM_{2.5} chemical components for two reasons: 1. It will dilute the focus of the current manuscript, which is the relationship between PM_{2.5} mass and OP (or cytotoxicity), 2. The relationship between toxicity and chemical components requires an in-depth discussion and we plan to prepare another manuscript on that topic, where we will include some mechanistic analyses as well along with statistical correlation. However, following the reviewer’s suggestion, we have added a brief discussion in the revised annotated manuscript about the association of different toxicity indicators and chemical species (Page 8, Line 161 to Page 9, Line 167). The discussion has been reproduced here for the convenience of the reviewer:

“A simple correlation analysis conducted between OP (or cytotoxicity) vs. measured chemical components (Supplementary Table 5 in SI), showed that different OP and cytotoxicity endpoints were associated with different chemical species in different regions. In general, OP^{OH_v} showed a

strong correlation with Fe, Cu, and WSOC ($r > 0.5$), OP^{DTT}_v was strongly correlated with Fe, Mn, and Cu ($r > 0.6$), and OP^C_v was associated with Co, Mn, Fe, and Cu ($r > 0.5$). OP^{GSH}_v showed moderate association with Cu, Al, and K, while CDv showed a strong correlation only with Fe and WSOC ($r > 0.5$).”

We have also added Supplementary Table 5 in SI showing the correlation coefficients (Pearson’s r) between extrinsic OP, CD, and different chemical species for different regions. The Table has been reproduced here for the convenience of the reviewer.

Supplementary Table 5: Pearson’s r for the correlation of ambient concentrations of different PM_{2.5} chemical species with extrinsic OP and cytotoxicity.

	Midwest US					West Midlands				
	CDv	OP ^{OH} _v	OP ^{GSH} _v	OP ^C _v	OP ^{DTT} _v	CDv	OP ^{OH} _v	OP ^{GSH} _v	OP ^C _v	OP ^{DTT} _v
Al	0.27	0.48	0.28	0.27	0.47	0.27	0.65	0.28	0.25	0.19
As	0.19	0.26	-0.05	0.25	0.12	0.46	0.29	0.28	0.51	0.35
Ba	0.22	0.18	-0.03	0.15	0.24	0.29	0.27	0.27	0.45	0.47
Cd	0.24	0.33	0.11	0.30	0.20	0.55	0.50	0.52	0.66	0.64
Co	0.25	0.23	-0.05	0.23	0.11	0.33	0.77	0.45	0.48	0.55
Cr	0.21	0.33	0.33	0.23	0.23	0.24	0.72	0.40	0.42	0.54
Cu	0.13	0.55	0.03	0.36	0.50	0.81	0.65	0.78	0.91	0.61
Fe	0.28	0.50	0.17	0.36	0.20	0.40	0.78	0.51	0.57	0.61
Ga	0.23	0.18	-0.04	0.15	0.24	0.31	0.31	0.29	0.46	0.48
K	0.18	0.17	0.02	0.12	0.24	0.38	0.31	0.30	0.55	0.44
Li	0.23	0.19	-0.16	0.35	0.21	0.66	0.65	0.69	0.76	0.70
Mn	0.31	0.29	-0.05	0.41	0.18	0.70	0.74	0.81	0.82	0.71
Ni	0.07	0.02	0.02	0.06	-0.03	0.48	0.69	0.52	0.56	0.57
Pb	0.16	0.30	0.25	0.17	-0.01	0.36	0.21	0.14	0.37	0.04
Rb	0.25	0.26	-0.01	0.34	0.38	0.66	0.77	0.67	0.82	0.62
Sr	0.18	0.16	0.00	0.12	0.22	0.20	0.08	0.06	0.32	0.22
V	0.06	0.06	-0.09	0.08	0.08	0.64	0.72	0.80	0.78	0.73
Zn	0.21	0.17	0.16	0.25	0.25	0.66	0.49	0.52	0.72	0.48
WSOC	0.13	0.04	-0.18	0.06	-0.04	-0.16	-0.21	-0.20	-0.14	-0.21

	Chile					Atlanta				
	CDv	OP ^{OH} _v	OP ^{GSH} _v	OP ^C _v	OP ^{DTT} _v	CDv	OP ^{OH} _v	OP ^{GSH} _v	OP ^C _v	OP ^{DTT} _v
Al	0.21	0.54	0.34	0.56	0.54	0.26	0.10	0.33	0.13	0.07
As	0.44	0.65	0.44	0.61	0.80	-0.10	-0.10	-0.25	-0.12	-0.03
Ba	0.28	0.40	0.42	0.36	0.53	-0.01	0.41	0.19	-0.21	0.21
Cd	0.25	0.31	0.44	0.65	0.75	-0.37	-0.09	-0.25	-0.44	-0.21
Co	0.14	0.25	0.21	0.32	0.39	0.17	0.24	0.26	0.17	0.06
Cr	0.13	0.23	0.24	0.29	0.34	0.19	0.23	0.17	0.27	0.07
Cu	0.37	0.43	0.43	0.56	0.75	0.47	0.63	-0.18	0.48	0.44
Fe	0.14	0.24	0.21	0.29	0.35	0.65	0.33	0.21	0.75	0.66
Ga	0.26	0.38	0.42	0.35	0.52	-0.03	0.35	0.23	-0.23	0.16
K	0.36	0.62	0.41	0.54	0.71	0.44	0.18	0.34	0.42	0.36
Li	0.34	0.48	0.24	0.56	0.69	0.03	0.18	0.15	-0.11	0.13
Mn	0.31	0.44	0.37	0.59	0.70	0.19	0.15	0.21	0.06	0.05
Ni	0.15	0.25	0.24	0.32	0.37	0.07	0.22	0.37	0.11	-0.06
Pb	0.39	0.47	0.49	0.59	0.73	0.08	0.24	-0.13	0.13	0.40
Rb	0.42	0.72	0.28	0.59	0.74	-0.29	0.11	0.11	-0.38	0.08
Sr	0.35	0.52	0.41	0.50	0.70	0.06	0.08	0.44	0.04	0.22
V	0.18	0.33	0.34	0.33	0.37	0.22	0.12	0.27	0.20	-0.03
Zn	0.31	0.47	0.50	0.48	0.63	0.55	0.37	0.28	0.30	0.22
WSOC	0.51	0.35	0.17	0.33	0.38	0.55	0.44	-0.08	0.33	0.09

	India				
	CDv	OP ^{OH} _v	OP ^{GSH} _v	OP ^C _v	OP ^{DTT} _v
Al	-0.18	-0.07	0.19	-0.24	-0.17
As	0.29	0.28	0.64	0.13	0.42
Ba	0.28	0.27	0.57	0.01	0.42
Cd	-0.08	-0.34	-0.27	0.18	0.11
Co	0.46	-0.14	0.27	0.66	0.84
Cr	0.21	0.00	0.10	0.05	0.10
Cu	-0.21	0.19	0.28	0.23	-0.13
Fe	0.24	-0.14	0.03	0.34	0.59
Ga	0.29	0.27	0.57	0.01	0.42
K	0.23	0.38	0.48	-0.11	0.14
Li	0.39	0.04	0.38	0.70	0.46
Mn	0.44	-0.07	0.24	0.73	0.65
Ni	-0.04	-0.02	-0.17	-0.11	0.003
Pb	-0.05	-0.02	-0.17	-0.12	-0.02
Rb	0.53	-0.06	0.51	0.41	0.80
Sr	0.37	0.20	0.27	0.31	0.69
V	0.50	0.16	-0.01	0.29	0.36
Zn	0.27	0.05	-0.09	0.26	0.25
WSOC	0.42	0.72	0.51	0.10	0.01

Reviewer #3 (Remarks to the Author):

Salana et al. present comprehensive aerosol mass, toxicity, and chemical composition measurements performed on a large set of aerosol filter samples collected over a broad geographical range. This is an important and valuable dataset because it allows for direct comparison between the samples collected in different locations, which is often not possible due to the non-standardized nature of the aerosol toxicity measurements performed by different research groups. This comparison allows the authors to reach the conclusion that the relationships between their measured health endpoints and PM mass is different in different locations. This is an important observation that has several interesting implications as discussed by the authors. I have a few specific comments that I believe require revision and/or clarification.

- 1) One important methodological detail that is currently missing is a description of how the PM_{2.5} mass concentrations were measured. Since part of the analysis hinges on the variability in PM_{2.5} mass concentrations relative to those in the measured health endpoints it would also be important to report some uncertainty estimates for the different measurements.**

Response:

We thank the reviewer for this comment. At most sites (except Atlanta), PM_{2.5} filters were gravimetrically weighed, and the collected PM_{2.5} mass was divided by total volume of sampled air to obtain the PM_{2.5} mass concentrations. At Atlanta, PM_{2.5} mass concentrations were measured using a tapered element oscillating microbalance (TEOM). We have now added this information in the subsection “Sampling site and sampling periods” (Page 19, Lines 396 to 398).

The uncertainties in PM_{2.5} mass concentrations in different regions varied between 7-10%. The details of the uncertainties for various instruments used for the collection and weighing of filters and the overall uncertainty in PM_{2.5} mass concentrations for various regions have now been provided in Supplementary Table 2 of the SI.

We also calculated the uncertainties in extrinsic OP and cytotoxicity measurements as well as the uncertainties in the ambient concentrations of different chemical species. Overall, the uncertainties in OP and cytotoxicity measurements ranged from 5% to 13%, while those in the ambient concentrations of chemical components ranged from 9% to 12%. All these uncertainties have now been added to Supplementary Table 2, which is reproduced here for the convenience of the reviewer.

Supplementary Table 2: Uncertainty (%) estimates for various measurements i.e., PM_{2.5} mass, chemical components, OP, and cytotoxicity*

Region	PM _{2.5} metric	Instrument used for the measurement	Total Uncertainty in the measurement (%)	
Midwest US	PM _{2.5} mass	Tisch Environmental Hi Vol Sampler for filter collection and Satorius, A120S for filter weighing	7.8	
	OP or Toxicity endpoints	Cytotoxicity	SpectraMax microplate reader (Molecular Devices, CA)	9.9
		OP ^{OH} _v	SAMERA	10.2
		OP ^{GSH} _v	SAMERA	11.8
		OP ^C _v	Bench-top spectrofluorometer (RF-5301 pc, Shimadzu Co., Japan)	6.9
		OP ^{DTT} _v	SAMERA	8.9
	Chemical Composition	Metals	NexION 300X ICP-MS; Perkin Elmer, Waltham, MA	10.0
WSOC		TOC Analyzer, Shimadzu Co., Japan	9.5	
West Midlands	PM _{2.5} mass	DIGITEL Hi Vol Sampler for filter collection and Sartorius, LE324S for filter weighing	9.9	
	OP or Toxicity endpoints	Cytotoxicity	SpectraMax microplate reader (Molecular Devices, CA)	9.3
		OP ^{OH} _v	SAMERA	11.3
		OP ^{GSH} _v	SAMERA	12.8
		OP ^C _v	Bench-top spectrofluorometer (RF-5301 pc, Shimadzu Co., Japan)	8.4
		OP ^{DTT} _v	SAMERA	10.2
	Chemical Composition	Metals	NexION 300X ICP-MS; Perkin Elmer, Waltham, MA	11.7
WSOC		TOC Analyzer, Shimadzu Co., Japan	11.3	
Chile	PM _{2.5} mass	MCV, S.A. Hi Vol Sampler model CAV-A /Mb for filter collection and Radwag XA 110/4Y for filter weighing	7.2	
	OP or Toxicity endpoints	Cytotoxicity	SpectraMax microplate reader (Molecular Devices, CA)	7.3
		OP ^{OH} _v	SAMERA	9.8
		OP ^{GSH} _v	SAMERA	11.5
		OP ^C _v	Bench-top spectrofluorometer (RF-5301 pc, Shimadzu Co., Japan)	6.2
		OP ^{DTT} _v	SAMERA	8.4
	Chemical Composition	Metals	NexION 300X ICP-MS; Perkin Elmer, Waltham, MA	9.6
WSOC		TOC Analyzer, Shimadzu Co., Japan	9.0	
Atlanta	PM _{2.5} mass	Thermo Anderson Hi Vol Sampler for filter collection and Tapered element oscillating microbalance (TEOM), Thermo Scientific TEOM 1400a for PM _{2.5} mass measurement	10.0	
	OP or Toxicity endpoints	Cytotoxicity	SpectraMax microplate reader (Molecular Devices, CA)	6.5
		OP ^{OH} _v	SAMERA	9.1
		OP ^{GSH} _v	SAMERA	10.9
		OP ^C _v	Bench-top spectrofluorometer (RF-5301 pc, Shimadzu Co., Japan)	5.1
		OP ^{DTT} _v	SAMERA	7.6
	Chemical Composition	Metals	NexION 300X ICP-MS; Perkin Elmer, Waltham, MA	11.8
WSOC		TOC Analyzer, Shimadzu Co., Japan	11.4	
India	PM _{2.5} mass	Thermo Scientific Hi Vol Sampler for filter collection and Sartorius, LA130S-F for filter weighing	10.3	
	OP or Toxicity endpoints	Cytotoxicity	SpectraMax microplate reader (Molecular Devices, CA)	6.7
		OP ^{OH} _v	SAMERA	9.3
		OP ^{GSH} _v	SAMERA	11.0
		OP ^C _v	Bench-top spectrofluorometer (RF-5301 pc, Shimadzu Co., Japan)	5.3
		OP ^{DTT} _v	SAMERA	7.8
	Chemical Composition	Metals	NexION 300X ICP-MS; Perkin Elmer, Waltham, MA	12.1
WSOC		TOC Analyzer, Shimadzu Co., Japan	11.6	

**Uncertainty in PM_{2.5} mass measurements (σ_m) was calculated as follows:*

$$\sigma_m = \sqrt{(\sigma_f)^2 + (\sigma_w)^2}$$

where, σ_f is the uncertainty in volume of air sampled (2-7%; depending on the sampler) and σ_w is the uncertainty in weighing filters (6-10%; depending on the weighing balance). At Atlanta site, uncertainty in PM_{2.5} mass measurements was the uncertainty of Tapered Element Oscillating Microbalance (TEOM) as provided by the manufacturer.

Uncertainties in OP_v and CD_v measurements (σ_{OP} and σ_{CD}) were calculated as follows:

$$\sigma_{OP}(\text{OR } \sigma_{CD}) = \sqrt{(\sigma_f)^2 + (\sigma_{me})^2 + (\sigma_e)^2}$$

where, σ_{me} is the uncertainty of the instrument used for the specific measurement and σ_e is the uncertainty in PM_{2.5} extraction (2%). The uncertainties (σ_{me}) for SAMERA, used for acellular OP [OP^{OH}_v (9%), OP^{GSH}_v (11%) and OP^{DTT}_v (7%)] measurements were calculated from accuracy and precision data reported in our previous publication³⁸. The uncertainties for the instruments used for measuring OP^C_v [Benchtop spectrofluorometer (4%)] and CD_v [SpectraMax microplate reader (6%)] were calculated from the accuracy and precision of the instruments provided by the manufacturer.

Uncertainty in the concentrations of chemical species (σ_c) was calculated as follows:

$$\sigma_c = \sqrt{(\sigma_m)^2 + (\sigma_e)^2 + (\sigma_i)^2}$$

where, σ_i the uncertainty of ICP-MS (6%), or TOC analyzer (5%) used for metals and WSOC measurements, respectively, as calculated from the accuracy and precision data of the instruments provided by the manufacturer.”

- 2) The focus of the paper is on the relationship between the 5 aerosol toxicity metrics and PM_{2.5} mass. In my opinion this is a good choice since a clear message emerges from the paper. Nevertheless, many experts will also be interested in the relationships between the 5 toxicity metrics themselves, since the development of such metrics is an active area of research. Perhaps the authors could add a heatmap or scatterplot matrix to the supplementary information to show these relationships, while noting that a detailed discussion of all the differences would be beyond the scope of the paper.**

Response:

We thank the reviewer for this suggestion. We have added a brief discussion and heatmap showing the correlation among different endpoints in SI of the revised manuscript. For the convenience of the reviewer, we have reproduced Supplementary Discussion 1 and Supplementary Fig. 9 below:

“Supplementary Discussion 1: Correlation among different OP and toxicity endpoints

We conducted a correlation analysis among all OP and cytotoxicity endpoints. The heatmap with the correlation coefficients (Pearson’s r) for different regions is shown in Supplementary Fig. 9. The figure shows that there was no appreciable correlation among most of the intrinsic endpoints,

except for the correlation between OP^Cm and CDm ($r > 0.5$). The correlations among extrinsic endpoints were better, although no consistent pattern was observed across different regions. For example, CDv showed a strong correlation with OP^Cv ($r > 0.6$) and a moderate correlation with almost all acellular OP ($0.4 < r < 0.6$) in most of the regions, except Midwest US. On the other hand, OP^{GSHv} and OP^{OHv} showed a weak correlation ($r < 0.4$) with other endpoints at most of the sites except West Midlands ($r > 0.4$). A more detailed discussion about these correlations and their mechanistic explanation is beyond the scope of the current paper.”

Supplementary Fig. 9: Heatmaps showing correlations (Pearson’s r) among different OP and cytotoxicity endpoints for different regions. Panel a) shows correlations among intrinsic endpoints and b) shows correlations between extrinsic endpoints.

3) L46: Typo? 'a dearth', rather than simply 'dearth'.

Response:

The typo has been corrected to “a dearth”.

4) L84: To perform a study with such broad geographical coverage it is necessary to use offline filter samples. However, this brings the limitation that the toxicity measurements will not capture short-lived ROS and OP-relevant species. I think it is worth briefly discussing this limitation, especially since the samples had variable storage times.

Response:

We thank the reviewer for this suggestion. We agree that due to the use of offline filter samples and variable storage times, short-lived ROS and some of the OP-relevant species might have been decayed. Previous studies have shown that short-lived compounds such as semi-volatile organic compounds could contribute significantly to OP^{DTT} and cellular OP^{57-59} . Moreover, studies investigating the $PM_{2.5}$ -bound ROS have shown that using offline filter methods to collect $PM_{2.5}$

might lead to the loss of several ROS species such as peroxy radicals and peroxide-containing highly oxygenated molecules (HOMs)⁶⁰⁻⁶². All these factors could lead to underestimation of OP and cytotoxicity of the PM_{2.5} samples collected in our study.

We have added the following sentences (Page 14, Line 292 to Page 15, Line 298) in the “Limitations and implications” subsection of the revised annotated manuscript to discuss this limitation:

“Although substantial efforts were resourced in our study to coordinate PM_{2.5} sampling in various parts of the world, it had some limitations which should be carefully considered before the general implication of our results. First, the collection and transport of filters from such an extensive spatial scale leads to unavoidable artifacts related to offline filter collection and the variable periods of sample storage, which could result in the loss of short-lived redox-active compounds, e.g., peroxy radicals and peroxide-containing highly oxygenated molecules (HOMs)⁶⁰⁻⁶², and semi-volatile organic compounds⁶⁰. Thus, the OP and cytotoxicity of the PM_{2.5} samples collected in our study could have been underestimated.”

- 5) **L131: Could these differences be partly due to the use of CoV as a metric, given this partially depends on sample size, which is more than an order of magnitude higher for the Midwest US than India? The authors might consider ways of demonstrating that sample size is not a critical factor when making comparisons across the different regions.**

Response:

We thank the reviewer for this comment. We agree that CoV partially depends on the sample size, and it could be an issue while comparing the CoVs of Midwest US with those of other regions like India, Atlanta, and West Midlands. Therefore, following the reviewer’s suggestion, we further checked if the differences in CoVs observed between different sites and endpoints are indeed significant. We used the asymptotic test for the equality of coefficients of variation as proposed by Feltz and Miller⁶³, using the R package, *cvequality*⁶⁴ to calculate the *p* values for site-wise and within-site comparison of CoVs for different endpoints. These *p* values are now added as Supplementary Table 3 in the SI of the revised manuscript.

As can be seen from this table, most of differences in the CoVs we discussed in the manuscript are indeed statistically significant ($p < 0.05$), despite the differences in their respective sample sizes. For example, the *p* values for the difference of CoV of PM_{2.5} mass in Midwest US vs. CoVs of PM_{2.5} mass at rest of the regions, was < 0.05 for all the cases, except for Atlanta. *p* values for the difference of CoV of PM_{2.5} mass in India vs. Midwest US was also significant ($p = 0.01$). Similarly, *p* values for the difference between CoVs of PM_{2.5} mass and the CoVs of extrinsic OP and cytotoxicity were < 0.05 for most of the sites. However, some of the differences were also non-significant (i.e., $p > 0.05$). In the revised manuscript, we have removed all the claims where the differences between respective CoVs were not significant. The manuscript is updated with the *p* values at all relevant places comparing CoVs of PM_{2.5} mass at different sites and the CoVs of two PM_{2.5} endpoints at a given site. Supplementary Table 3 is reproduced here for the convenience of the reviewer.

Supplementary Table 3: *p* values for the site-wise and within-site comparison of CoVs for different PM_{2.5} endpoints. *p* values ≤ 0.05 are shown in bold.

	Endpoint	Site	Midwest US	West Midlands	Chile	Atlanta	India
p values for comparison of CoVs for different PM_{2.5} endpoints across different regions	Mass	Midwest US		0.04	<0.001	0.44	0.01
		West Midlands			0.07	0.48	0.74
		Chile				0.02	0.15
		Atlanta					0.31
	CDv	Midwest US		0.02	0.01	0.52	0.98
		West Midlands			0.67	0.10	0.25
		Chile				0.16	0.35
		Atlanta					0.63
	OP ^{OH} _v	Midwest US		0.64	0.03	0.04	0.41
		West Midlands			0.24	0.05	0.71
		Chile				0.02	0.18
		Atlanta					0.11
	OP ^{GSH} _v	Midwest US		0.45	0.07	0.41	0.34
		West Midlands			0.83	0.92	0.71
		Chile				0.95	0.79
		Atlanta					0.79
	OP ^C _v	Midwest US		0.28	0.71	0.16	0.84
		West Midlands			0.46	0.06	0.43
		Chile				0.13	0.72
		Atlanta					0.26
OP ^{DTT} _v	Midwest US		0.57	<0.001	0.28	0.04	
	West Midlands			0.12	0.75	0.39	
	Chile				0.19	0.46	
	Atlanta					0.57	
p values for comparison of CoV of Mass vs OPv (and CDv)	CDv		<0.001	0.004	0.06	0.04	0.11
	OP ^{OH} _v		<0.001	0.03	0.01	0.30	0.15
	OP ^{GSH} _v		<0.001	0.005	0.05	0.002	0.03
	OP ^C _v		<0.001	0.01	0.50	0.15	0.14
	OP ^{DTT} _v		<0.001	0.20	0.10	0.04	0.10

The modified text (Page 25, Line 548 to Page 26, Line 551 of the revised annotated manuscript) is reproduced here for the convenience of the reviewer.

“Statistical significance of the differences in CoVs observed between different sites and OP (or cytotoxicity) endpoints was determined by the asymptotic test for the equality of CoVs as proposed by Feltz and Miller⁶³ using the R package, cvequality (Version 0.1.3)⁶⁴. ”

And Page 7, Line 139 to Page 8, Line 150 of the revised annotated manuscript:

“Interestingly, despite the longest sampling span in the Midwest US (one year), the variation in PM_{2.5} mass concentrations (CoV = 32%; Figure 2) was lower (*p* < 0.05; see Supplementary Table 3 in SI for statistical significance of the differences in CoVs observed between different sites and endpoints) than in most of the regions except Atlanta. The variation in PM_{2.5} mass concentrations in India was significantly larger (CoV = 49%), compared to the Midwest US (*p* = 0.01) despite only 17 samples collected from India. CoVs of PM_{2.5} mass in Chile (CoV = 77%) were higher than in the Midwest US (*p* < 0.001) and Atlanta (*p* = 0.02).

In general, the variations in PM_{2.5} mass were significantly lower than that in OP_v and CD_v at most sites with few exceptions (see Supplementary Table 3), which was supported by all three metrics used to assess variability, i.e., CoV, RCV_Q, and RCV_M.”

- 6) **L138: One other potential issue with the use of CoV as a statistical metric is inflated values due to division by means that approach zero. I am wondering if this could be an issue for some of the GSH_v CoV values, given the GSH_v measurements sometimes approach zero, and even more so for the element CoV values shown in Fig. S7, since I assume some of the elements were present in very low concentrations. The authors could look at additional metrics of variability to support their observation that the elemental concentrations varied more widely than the corresponding total mass concentrations.**

Response:

We thank the reviewer for this comment. We agree that CoV based on mean values may not be reliable in the cases when mean values approach zero. This is true for some of the endpoints used in our study such as OP^{GSH_v} and the ambient concentrations of elements. To address this issue, we have calculated two more metrics of variability — robust coefficient of variation based on interquartile range (RCV_Q) and robust coefficient of variation based on median (RCV_M)⁶⁵, to support our observation that the variations in total PM_{2.5} mass concentrations were lower than the concentrations of individual PM_{2.5} chemical species.

$RCV_Q = (0.75 \times IQR)/m$; where IQR = interquartile range and m is the sample median

$RCV_M = (1.4826 \times MAD)/m$; where MAD = Median Absolute Deviation and m is the sample median

Comparison of RCV_Q and RCV_M values for PM_{2.5} mass concentrations with those for the OP_v, CD_v and ambient concentrations of all the chemical species, shows that the variations in the mass concentrations are in general lower irrespective of the metric used.

We have now revised the figure in SI with a heatmap showing the three different metrics of variability for various extrinsic PM_{2.5} endpoints, i.e., mass, volume normalized acellular and cellular OP, CD, and ambient concentrations of various PM_{2.5} chemical species (µg/m³) at different sampling sites. We have also added a few sentences in the revised annotated manuscript (Page 7, Lines 133 to 139) to include the discussion on these additional parameters we calculated to represent the variability of different endpoints. Both the figure and the revised discussion are reproduced here for the convenience of the reviewer:

“Since CoV is more sensitive to the outliers and can be inflated in the cases when arithmetic averages of the data approaches zero, we also quantified the variations in different endpoints using two more metrics: robust coefficient of variation based on interquartile range [$RCV_Q = 0.75 \times (\text{interquartile range} \times 100)/\text{median}$] and robust coefficient of variation based on median [$RCV_M = 1.483 \times (\text{median absolute deviation} \times 100)/\text{median}$]⁶⁵. The RCV_Q and RCV_M values for different extrinsic endpoints (i.e., PM_{2.5} mass, OP_v and CD_v) at various sites are given in Supplementary Fig. 8.”

Page 8, Lines 147 to 150 of the revised annotated manuscript:

“In general, the variations in $PM_{2.5}$ mass were significantly lower than that in OP_v and CD_v at most sites with few exceptions (see Supplementary Table 3), which was supported by all three metrics used to assess variability, i.e., CoV , RCV_Q , and RCV_M .”

Page 8, Lines 160 to 161 of the revised annotated manuscript:

“We also calculated the CoV , RCV_Q and RCV_M for the chemical components measured in our study (Supplementary Fig. 8 in SI).”

And Page 9, Lines 181 to 182 of the revised annotated manuscript:

“Both RCV_Q and RCV_M also showed a similar trend as CoV , i.e., higher values for chemical components than $PM_{2.5}$ mass.”

Supplementary Fig. 8: Heatmap showing three different metrics of variability [CoV , RCV_Q and RCV_M (%)] for various extrinsic $PM_{2.5}$ endpoints, i.e., mass, volume normalized acellular and cellular OP , and CD , and ambient concentrations of various $PM_{2.5}$ chemical species ($\mu\text{g}/\text{m}^3$) at individual sampling sites. RCV_Q represents robust coefficient of variation based on interquartile range and RCV_M represents robust coefficient of variation based on the median. For India, we have clubbed all the sites together because the number of $PM_{2.5}$ samples from each site is much lower ($n < 5$) as compared to all other sites ($n > 10$) (Ahmedabad, $n = 3$; Patiala, $n = 5$; Hisar, $n = 5$; Faridabad, $n = 5$).

- 7) **L151:** For completeness it should be mentioned that there are chemical species that were not resolved by the measurements that could be influencing the observed variability in the measured health endpoints (e.g. different organic aerosol fractions within the total WSOC).

Response:

We thank the reviewer for this suggestion. We have added the following sentences on Page 9, Lines 167 to 170 of the revised annotated manuscript to highlight this limitation:

“Note, WSOC is a bulk species containing a variety of organic compounds, such as polycyclic aromatic hydrocarbons (PAHs), quinones, carboxylic acids, aldehydes, and amides⁴⁴, and measuring the composition of organic aerosols at such a chemically resolved scale is beyond the scope of our current study.”

- 8) **L215:** The non-linearity is mainly driven by the 1 set of India samples with generally lower intrinsic toxicity. To some extent this is an unavoidable limitation of the sample size (385 samples is already a lot relative to other studies). Nevertheless I think it argues against the usefulness of purely data-driven non-linear fits. For example, do the authors consider the cubic fits or CDv and OHv against PM_{2.5} to be physically reasonable? What are the risks of extrapolating these relationships beyond their fitted ranges? I wonder if a better approach would be to assume a specific non-linear functional form (e.g. a log function) that has some basis in the literature and/or theory, and then to compare that against a standard linear fit. Are the authors aware of any non-linear functional forms that have been proposed in previous studies that could perform this role?

Response:

We thank the reviewer for this suggestion. As suggested by the reviewer, we tried fitting other non-linear functional forms (e.g., piecewise linear functions, cubic spline, Poisson regression and logistic regression functions) used in previous studies and found that the logistic regression curve fits our data best. Logistic regression has been used in several epidemiological studies exploring the relationships between PM_{2.5} mass and health effects such as the relationship between exposure to PM_{2.5} mass and under-5 mortality in China⁶⁶, all-cause mortality in the US⁶⁷, asthma morbidity in rural USA⁶⁸, acute myocardial infarction in USA⁶⁹, and elevated platelet counts in Taiwanese adults⁷⁰. We fitted logistic regression curve to all endpoints using the Python library called SciPy. We found that logistic regression curves perform better than the standard linear fits for all the endpoints used in our study. Furthermore, to demonstrate that the logistic regression curve provides physically relevant estimations even after extrapolating for PM_{2.5} concentrations beyond the fitted range, we have plotted the curves for concentrations up to 700 $\mu\text{g}/\text{m}^3$ (these are included only in this response document; see figure R1). Interestingly, these curves plotted beyond the collected data range, still remained consistent with the conclusions made earlier (i.e., within the collected data range) regarding a steeper slope at lower PM_{2.5} mass ($< 50 \mu\text{g}/\text{m}^3$) and gradual flattening at higher mass concentrations ($> 300 \mu\text{g}/\text{m}^3$).

We have now modified Figure 4 to replace the polynomial curves with the logistic regression curves. We have also added a brief discussion on the logistic regression curves in the revised

manuscript as well as the details of various parameters and Python package used to fit the logistic regression curve in SI. The modified Figure 4, and the related discussion in the main manuscript and SI are reproduced below for the convenience of the reviewer.

Figure 4: Relationship between extrinsic OP (and cytotoxicity) vs. PM_{2.5} mass concentrations based on the entire dataset plotted together. Here, the intrinsic OP and cytotoxicity are represented by the size of the bubble. Curves are fitted for a) the seasonally averaged data for all five regions considered in this study (n = 12); and b) entire dataset for all sites: OP^{DTTv} (n = 382), OP^{Cv} (n = 375), OP^{GSHv} (n = 385), OP^{OHv} (n = 380) and CDv (n = 365). In panel a, the blue line represents the linear curve fitted for all five regions and the orange line represents the linear curve after excluding India. In panel b, the blue line represents a linear curve, and the red line represents the fitted logistic curve. RMSE represents root mean squared error.

Page 11, Lines 208 to 217 of the revised annotated manuscript:

“Figure 4a shows the linear regression analysis of extrinsic OP and cytotoxicity vs. PM_{2.5} mass for the regional averages, while Figure 4b shows the same regression plot for the entire dataset using both linear and non-linear regression curves. We used logistic regression to model the non-linear relationship between OP_v (and CD_v) vs. PM_{2.5} mass. Logistic regression has been extensively used in several epidemiological studies to explain the relationships between PM_{2.5} mass and health effects, such as the relationship between exposure to PM_{2.5} and under-5 mortality in China⁶⁶, all-cause mortality in the US⁶⁷, asthma morbidity in rural USA⁶⁸, acute myocardial infarction in USA⁶⁹ and elevated platelet counts in Taiwanese adults⁷⁰. Details about various parameters and the software package used to fit the logistic regression curve are given in SI (see Supplementary Method 6).”

“Supplementary Method 6: Details regarding the logistic curves to explain the relationship between extrinsic OP (and cytotoxicity) vs. PM_{2.5} mass concentrations.

We used the optimization package from an open-source Python library called SciPy which is commonly used for solving scientific and technical computing problems such as optimization, integration, linear algebra and constructing special functions. A logistic function is defined as:

$$Y = \frac{L}{1 + e^{(-k(x-x_0))}} + b$$

Where,

x_0 is the value of the midpoint of the logistic function.

k = the logistic growth rate of the function

L = the maximum value of the Y in each iteration

b = bias in the function

Note, x is the dataset containing PM_{2.5} mass concentrations and Y is the predicted value of OP or cytotoxicity.

The initial guesses for the different parameters were as follows:

L = maximum value in y dataset [i.e., maximum value in the OP_v (or CD_v) dataset]

k = 0.1 for OP^{GSH}_v and 1 for all others

x_0 = median of x dataset (i.e., median of PM_{2.5} mass concentrations)

b = minimum value in y dataset

number of iterations = 4000”

Figure R1: Relationship between extrinsic OP (and cytotoxicity) vs. $PM_{2.5}$ mass concentrations for $PM_{2.5}$ concentrations up to $700 \mu g/m^3$. Here, the intrinsic OP and cytotoxicity are represented by the size of the bubble. The blue line represents a linear curve, and the red line represents the fitted logistic curve.

- 9) **Related to the above point, a clear point that emerges from Fig. 4 is the need for more measurements at medium and high PM concentrations to better constrain the non-proportional relationships between these metrics and PM_{2.5} mass. Perhaps the authors could discuss this point as a goal for future studies.**

Response:

We thank the reviewer for this suggestion. We agree that more measurements at medium and high PM_{2.5} concentrations could better define the PM_{2.5} vs. OP (and cytotoxicity) curves. We have added the following sentences in the “Limitations and implications” subsection of our revised annotated manuscript to highlight this limitation (Page 16, Lines 320 to 326):

“Although, the overall range of PM_{2.5} mass concentrations obtained from our samples is quite large (2-561 µg/m³), it is still mostly dominated by the samples with mass concentrations < 50 µg/m³ (354 out of 385 samples), and with only 19 samples having mass concentrations in the range of 50-200 µg/m³. Thus, the curves shown in Figure 4 could be somewhat biased by the samples with low PM_{2.5} mass concentrations. Future studies should focus on more measurements at medium and high PM_{2.5} concentrations to better constrain the non-proportional relationships between health metrics and PM_{2.5} mass.”

- 10) **Figure 4: The smallest data points (lowest intrinsic values) are difficult to see. The size of these data points could be increased for better visualization, perhaps while reducing the number of intrinsic categories (e.g. from >6 to ~3 or 4) so that the largest marker size is not too large.**

Response:

Figure 4 has now been modified by reducing the intrinsic categories to increase the relative size of the data points. The modified figure has been reproduced here for the convenience of the reviewer. We have reproduced the figure as a response to a previous comment (comment #8) by the reviewer. The same figure has also been added below.

Figure 4: Relationship between extrinsic OP (and cytotoxicity) vs. $PM_{2.5}$ mass concentrations based on the entire dataset plotted together. Here, the intrinsic OP and cytotoxicity are represented by the size of the bubble. Curves are fitted for a) the seasonally averaged data for all five regions considered in this study (n = 12); and b) the entire dataset for all sites: OP^{DTTv} (n = 382), OP^{Cv} (n = 375), OP^{GSHv} (n = 385), OP^{OHv} (n = 380) and CDv (n = 365). In panel a, the blue line represents the linear curve fitted for all five regions and the orange line represents the linear curve after excluding India. In panel b, the blue line represents a linear curve, and the red line represents the fitted logistic curve. RMSE represents root mean squared error.

REFERENCES

1. Gao, D., Fang, T., Verma, V., Zeng, L. & Weber, R. J. A method for measuring total aerosol oxidative potential (OP) with the dithiothreitol (DTT) assay and comparisons between an urban and roadside site of water-soluble and total OP. *Atmos. Meas. Tech.* **10**, 2821–2835 (2017).
2. Roper, C., Delgado, L. S., Barrett, D., Massey Simonich, S. L. & Tanguay, R. L. PM 2.5 Filter Extraction Methods: Implications for Chemical and Toxicological Analyses. *Environ. Sci. Technol.* **53**, 434–442 (2019).
3. Bein, K. J. & Wexler, A. S. A high-efficiency, low-bias method for extracting particulate matter from filter and impactor substrates. *Atmos. Environ.* **90**, 87–95 (2014).
4. Zhang, W., Yu, H., Hettiyadura, A. P. S., Verma, V. & Laskin, A. Field evidence for enhanced generation of reactive oxygen species in atmospheric aerosol containing quinoline components. *Atmos. Environ.* **291**, 119406 (2022).
5. Xu, Z. *et al.* Potential underestimation of ambient brown carbon absorption based on the methanol extraction method and its impacts on source analysis. *Atmos. Chem. Phys.* **22**, 13739–13752 (2022).
6. Godri, K. J. *et al.* Particulate matter oxidative potential from waste transfer station activity. *Environ. Health Perspect.* **118**, 493–498 (2010).
7. Yang, A. *et al.* Measurement of the oxidative potential of PM_{2.5} and its constituents: The effect of extraction solvent and filter type. *Atmos. Environ.* **83**, 35–42 (2014).
8. Roper, C. *et al.* Characterization of ambient and extracted PM_{2.5} collected on filters for toxicology applications. *Inhal. Toxicol.* **27**, 673–681 (2016).
9. Farahani, V. J. *et al.* The oxidative potential of particulate matter (PM) in different regions around the world and its relation to air pollution sources. *Environ. Sci. Atmos.* **2**, 1076–1086 (2022).
10. Zhang, Z. H. *et al.* Are reactive oxygen species (ROS) a suitable metric to predict toxicity of carbonaceous aerosol particles? *Atmos. Chem. Phys.* **22**, 1793–1809 (2022).
11. Saffari, A., Daher, N., Shafer, M. M., Schauer, J. J. & Sioutas, C. Global perspective on the oxidative potential of airborne particulate matter: A synthesis of research findings. *Environ. Sci. Technol.* **48**, 7576–7583 (2014).
12. Verma, V. *et al.* Organic aerosols associated with the generation of reactive oxygen species (ROS) by water-soluble PM_{2.5}. *Environ. Sci. Technol.* **49**, 4646–4656 (2015).
13. Fang, T. *et al.* Oxidative potential of ambient water-soluble PM_{2.5} in the southeastern United States: Contrasts in sources and health associations between ascorbic acid (AA) and dithiothreitol (DTT) assays. *Atmos. Chem. Phys.* **16**, 3865–3879 (2016).
14. Saffari, A., Daher, N., Shafer, M. M., Schauer, J. J. & Sioutas, C. Seasonal and spatial variation in dithiothreitol (DTT) activity of quasi-ultrafine particles in the Los Angeles Basin and its association with chemical species. *J. Environ. Sci. Heal. - Part A* **49**, 441–

- 451 (2014).
15. Bates, J. T. *et al.* Reactive Oxygen Species Generation Linked to Sources of Atmospheric Particulate Matter and Cardiorespiratory Effects. *Environ. Sci. Technol.* **49**, 13605–13612 (2015).
 16. Abrams, J. Y. *et al.* Associations between Ambient Fine Particulate Oxidative Potential and Cardiorespiratory Emergency Department Visits. *Environ. Health Perspect.* **125**, 107008-1-107008–9 (2017).
 17. Yu, H., Puthussery, J. V., Wang, Y. & Verma, V. Spatiotemporal variability in the oxidative potential of ambient fine particulate matter in the Midwestern United States. *Atmos. Chem. Phys.* **21**, 16363–16386 (2021).
 18. Kumagai, Y. *et al.* Oxidation of proximal protein sulfhydryls by phenanthraquinone, a component of diesel exhaust particles. *Chem. Res. Toxicol.* **15**, 483–489 (2002).
 19. Cho, A. K. *et al.* Redox activity of airborne particulate matter at different sites in the Los Angeles Basin. *Environ. Res.* **99**, 40–47 (2005).
 20. Taylor, J. P. & Tse, H. M. The role of NADPH oxidases in infectious and inflammatory diseases. *Redox Biol.* **48**, 102159 (2021).
 21. Forman, H. J., Zhang, H. & Rinna, A. Glutathione: Overview of its protective roles, measurement, and biosynthesis. *Mol. Aspects Med.* **30**, 1–12 (2009).
 22. SIES, H. Strategies of antioxidant defense. *Eur. J. Biochem.* **215**, 213–219 (1993).
 23. Pham-Huy, L. A., He, H. & Pham-Huy, C. Free Radicals, Antioxidants in Disease and Health. *Int. J. Biomed. Sci.* **4**, 89–96 (2008).
 24. Akhtar, M. J. *et al.* Nanotoxicity of pure silica mediated through oxidant generation rather than glutathione depletion in human lung epithelial cells. *Toxicology* **276**, 95–102 (2010).
 25. Wang, Y. *et al.* Arsenic induces mitochondria-dependent apoptosis by reactive oxygen species generation rather than glutathione depletion in Chang human hepatocytes. *Arch. Toxicol.* **83**, 899–908 (2009).
 26. Han, Y. H., Kim, S. Z., Kim, S. H. & Park, W. H. Apoptosis in pyrogallol-treated Calu-6 cells is correlated with the changes of intracellular GSH levels rather than ROS levels. *Lung Cancer* **59**, 301–314 (2008).
 27. Gao, D., J. Godri Pollitt, K., A. Mulholland, J., G. Russell, A. & J. Weber, R. Characterization and comparison of PM_{2.5} oxidative potential assessed by two acellular assays. *Atmos. Chem. Phys.* **20**, 5197–5210 (2020).
 28. Shen, J. *et al.* Aerosol Oxidative Potential in the Greater Los Angeles Area: Source Apportionment and Associations with Socioeconomic Position. *Environ. Sci. Technol.* **56**, 17795–17804 (2022).
 29. Wang, Y., Plewa, M. J., Mukherjee, U. K. & Verma, V. Assessing the cytotoxicity of ambient particulate matter (PM) using Chinese hamster ovary (CHO) cells and its relationship with the PM chemical composition and oxidative potential. *Atmos. Environ.*

- 179, 132–141 (2018).
30. Wang, Y. *et al.* Sources of cellular oxidative potential of water-soluble fine ambient particulate matter in the Midwestern United States. *J. Hazard. Mater.* **425**, 127777 (2022).
 31. Mousavi, A. *et al.* Impact of emissions from the Ports of Los Angeles and Long Beach on the oxidative potential of ambient PM_{0.25} measured across the Los Angeles County. *Sci. Total Environ.* **651**, 638–647 (2019).
 32. Jiang, H., Jang, M., Sabo-Attwood, T. & Robinson, S. E. Oxidative potential of secondary organic aerosols produced from photooxidation of different hydrocarbons using outdoor chamber under ambient sunlight. *Atmos. Environ.* **131**, 382–389 (2016).
 33. Charrier, J. G. & Anastasio, C. Rates of Hydroxyl Radical Production from Transition Metals and Quinones in a Surrogate Lung Fluid. *Environ. Sci. Technol.* **49**, 9317–9325 (2015).
 34. Weichenthal, S. *et al.* Oxidative burden of fine particulate air pollution and risk of cause-specific mortality in the Canadian Census Health and Environment Cohort (CanCHEC). *Environ. Res.* **146**, 92–99 (2016).
 35. Weichenthal, S., Lavigne, E., Evans, G., Pollitt, K. & Burnett, R. T. Ambient PM_{2.5} and risk of emergency room visits for myocardial infarction: Impact of regional PM_{2.5} oxidative potential: A case-crossover study. *Environ. Heal. A Glob. Access Sci. Source* **15**, 1–9 (2016).
 36. Weichenthal, S. *et al.* Daily Summer Temperatures and Hospitalization for Acute Cardiovascular Events: Impact of Outdoor PM_{2.5} Oxidative Potential on Observed Associations Across Canada. *Epidemiology* **34**, 897–905 (2023).
 37. Daellenbach, K. R. *et al.* Sources of particulate-matter air pollution and its oxidative potential in Europe. *Nature* **587**, 414–419 (2020).
 38. Yu, H., Puthussery, J. V. & Verma, V. A semi-automated multi-endpoint reactive oxygen species activity analyzer (SAMERA) for measuring the oxidative potential of ambient PM_{2.5} aqueous extracts. *Aerosol Sci. Technol.* **54**, 304–320 (2019).
 39. Xiong, Q., Yu, H., Wang, R., Wei, J. & Verma, V. Rethinking Dithiothreitol-Based Particulate Matter Oxidative Potential: Measuring Dithiothreitol Consumption versus Reactive Oxygen Species Generation. *Environ. Sci. Technol.* **51**, 6507–6514 (2017).
 40. Reiniers, M. J. *et al.* Preparation and Practical Applications of 2',7'-Dichlorodihydrofluorescein in Redox Assays. *Anal. Chem.* **89**, 3853–3857 (2017).
 41. Zhang, Q. *et al.* ACSL1-induced ferroptosis and platinum resistance in ovarian cancer by increasing FSP1 N-myristylation and stability. *Cell Death Discov.* **9**, 1–12 (2023).
 42. Chen, W. Q., Lian, W. S., Yuan, Y. F. & Li, M. Q. The synergistic effects of oxaliplatin and piperlongumine on colorectal cancer are mediated by oxidative stress. *Cell Death Dis.* **10**, (2019).
 43. Yadav, R., Munan, S., Kardam, V., Dutta Dubey, K. & Samanta, A. Esterase Specific

- Fluorescent Probe: Mechanistic Understanding Using QM/MM Calculation and Cell States Discrimination. *Chem. - A Eur. J.* **29**, (2023).
44. Tang, H. *et al.* Graphene quantum dots obstruct the membrane axis of Alzheimer's amyloid beta. *Phys. Chem. Chem. Phys.* **24**, 86–97 (2022).
 45. Nelson, M. T. *et al.* Examining cellular responses to reconstituted antibody protein liquids. *Sci. Rep.* **11**, 1–9 (2021).
 46. Cooper, J. R. *et al.* Long term culture of the A549 cancer cell line promotes multilamellar body formation and differentiation towards an alveolar type II Pneumocyte phenotype. *PLoS One* **11**, 1–20 (2016).
 47. Chuquimia, O. D., Petursdottir, D. H., Periolo, N. & Fernández, C. Alveolar epithelial cells are critical in protection of the respiratory tract by secretion of factors able to modulate the activity of pulmonary macrophages and directly control bacterial growth. *Infect. Immun.* **81**, 381–389 (2013).
 48. Chan, M. & Liu, Y. Function of epithelial stem cell in the repair of alveolar injury. *Stem Cell Res. Ther.* **13**, 1–8 (2022).
 49. Feng, S. *et al.* The pathophysiological and molecular mechanisms of atmospheric PM_{2.5} affecting cardiovascular health: A review. *Ecotoxicol. Environ. Saf.* **249**, 114444 (2023).
 50. Du, Y., Xu, X., Chu, M., Guo, Y. & Wang, J. Air particulate matter and cardiovascular disease: The epidemiological, biomedical and clinical evidence. *J. Thorac. Dis.* **8**, E8–E19 (2016).
 51. Khajeh-Hosseini-Dalasm, N. & Longest, P. W. Deposition of Particles in the Alveolar Airways: Inhalation and Breath-Hold with Pharmaceutical Aerosols. *J. Aerosol Sci.* **79**, 15–20 (2015).
 52. Tian, H. *et al.* Comparative Ligandomics Analysis of Human Lung Epithelial Cells Exposed to PM_{2.5}. *Biomed. Environ. Sci.* **33**, 165–173 (2020).
 53. Vuong, N. Q. *et al.* In vitro toxicoproteomic analysis of A549 human lung epithelial cells exposed to urban air particulate matter and its water-soluble and insoluble fractions. *Part. Fibre Toxicol.* **14**, 1–19 (2017).
 54. Cheng, W. *et al.* Inhibition of inflammation-induced injury and cell migration by coelomin and militarine in PM_{2.5}-exposed human lung alveolar epithelial A549 cells. *Eur. J. Pharmacol.* **896**, 173931 (2021).
 55. Leibrock, L., Wagener, S., Singh, A. V., Laux, P. & Luch, A. Nanoparticle induced barrier function assessment at liquid-liquid and air-liquid interface in novel human lung epithelia cell lines. *Toxicol. Res. (Camb)*. **8**, 1016–1027 (2019).
 56. Wang, J. *et al.* Urban particulate matter triggers lung inflammation via the ROS-MAPK-NF- κ B signaling pathway. *J. Thorac. Dis.* **9**, 4398–4412 (2017).
 57. Pirhadi, M., Mousavi, A., Taghvaei, S., Shafer, M. M. & Sioutas, C. Semi-volatile components of PM_{2.5} in an urban environment: Volatility profiles and associated

- oxidative potential. *Atmos. Environ.* **223**, 117197 (2020).
58. Biswas, S. *et al.* Oxidative potential of semi-volatile and non volatile particulate matter (PM) from heavy-duty vehicles retrofitted with emission control technologies. *Environ. Sci. Technol.* **43**, 3905–3912 (2009).
 59. Vreeland, H. *et al.* Oxidative potential of PM_{2.5} during Atlanta rush hour: Measurements of in-vehicle dithiothreitol (DTT) activity. *Atmos. Environ.* **165**, 169–178 (2017).
 60. Krapf, M. *et al.* Labile Peroxides in Secondary Organic Aerosol. *Chem* **1**, 603–616 (2016).
 61. Fuller, S. J., Wragg, F. P. H., Nutter, J. & Kalberer, M. Comparison of on-line and off-line methods to quantify reactive oxygen species (ROS) in atmospheric aerosols. *Atmos. Environ.* **92**, 97–103 (2014).
 62. Zhou, J. *et al.* Predominance of secondary organic aerosol to particle-bound reactive oxygen species activity in fine ambient aerosol. *Atmos. Chem. Phys.* **19**, 14703–14720 (2019).
 63. Feltz, C. J. & Miller, G. E. An asymptotic test for the equality of coefficients of variation from k populations. *Stat. Med.* **15**, 647–658 (1996).
 64. Marwick, B. & Krishnamoorthy, K. cvequality: Tests for the Equality of Coefficients of Variation from Multiple Groups. R software package version 0.1.3. (2019).
 65. Arachchige, C. N. P. G., Prendergast, L. A. & Staudte, R. G. Robust analogs to the coefficient of variation. *J. Appl. Stat.* **49**, 268–290 (2022).
 66. He, C. *et al.* Fine particulate matter air pollution and under-5 children mortality in China: A national time-stratified case-crossover study. *Environ. Int.* **159**, 107022 (2022).
 67. Franklin, M., Zeka, A. & Schwartz, J. Association between PM_{2.5} and all-cause and specific-cause mortality in 27 US communities. *J. Expo. Sci. Environ. Epidemiol.* **17**, 279–287 (2007).
 68. Loftus, C. *et al.* Regional PM_{2.5} and asthma morbidity in an agricultural community: A panel study. *Environ. Res.* **136**, 505–512 (2015).
 69. Madrigano, J. *et al.* Long-term exposure to PM_{2.5} and incidence of acute myocardial infarction. *Environ. Health Perspect.* **121**, 192–196 (2013).
 70. Zhang, Z. *et al.* Long-term exposure to ambient particulate matter (PM_{2.5}) is associated with platelet counts in adults. *Environ. Pollut.* **240**, 432–439 (2018).

Reviewer #3 (Remarks to the Author):

The authors have responded effectively to all of my concerns and I have no further comments. I recommend publication.